# IS OPTIMAL TRANSPORT NECESSARY FOR INVERSE REINFORCEMENT LEARNING?

## ABSTRACT

Inverse Reinforcement Learning (IRL) aims to recover a reward function from expert demonstrations. Recently, Optimal Transport (OT) methods have been successfully deployed to align trajectories and infer rewards. While OT-based methods have shown strong empirical results, they are conceptually complicated, requiring the solution of the OT optimization problem. In this work, we challenge the necessity of OT in IRL by proposing two simple, heuristic alternatives: (1) Minimum-Distance Reward, which assigns rewards based on the nearest expert state regardless of temporal order; and (2) Segment-Matching Reward, which incorporates lightweight temporal alignment by matching agent states to corresponding segments in the expert trajectory. These methods circumvent optimization, exhibit linear-time complexity, and are easy to implement. Through extensive evaluations across 32 online and offline benchmarks with three reinforcement learning algorithms, we show that our simple rewards match or outperform recent OT-based approaches. Our findings suggest that the core benefits of OT may arise from basic proximity alignment rather than its optimal coupling formulation, advocating for reevaluation of optimal transport in future IRL design.

## 1 INTRODUCTION

Inverse Reinforcement Learning (IRL) addresses the problem of recovering a reward function from expert demonstrations, such that the expert behavior is (approximately) optimal under the learned rewards (Ng & Russell, 2000). This reward function can then be used to train agents that imitate the expert, even when explicit task rewards are unavailable. IRL is especially useful in applications where reward engineering is challenging or infeasible, such as robotics, healthcare, and human behavior modeling.

A recent line of work proposes to cast the IRL reward labeling problem as one of trajectory alignment via Optimal Transport (OT) (Luo et al., 2023; Fu et al., 2024). In the offline version of the problem setting, the dataset consists of expert and non-expert trajectories, none of which are labeled with rewards. Recent OT-based methods estimate the underlying rewards for all the states in the dataset, and then apply an offline RL algorithm to find a policy. To estimate the rewards, the approach defines a cost between states, computes a coupling between expert and non-expert trajectories that minimizes the transport cost, and uses the resulting alignment to define proxy rewards. This strategy has demonstrated success across various offline and online RL benchmarks, and has motivated further refinements such as temporal masking, context-aware distances, and alignment constraints (Wang et al., 2024a; Bobrin et al., 2024).

Despite their empirical promise, OT-based methods approximate the proximity between agent and expert trajectories by solving the OT optimization problem (e.g., via Sinkhorn iterations). Such a black-box inductive bias raises a natural but underexplored question:

*Is optimal transport actually necessary for IRL?* Or, put differently, can we match its performance using simpler, interpretable heuristic alternatives that avoid solving any optimization problem?

In this paper, we take a minimalist view of reward design for IRL. Specifically, we propose two simple and interpretable reward functions: *Minimum-Distance Reward*, which assigns each non-expert state a reward based on its closest distance to the expert trajectory, ignoring any notion of time or ordering; and *Segment-matching Reward*, which introduces a lightweight temporal structure by dividing the

expert trajectory into segments and assigning rewards based on proximity to temporally aligned expert segments. These two reward functions are conceptually simple, require no learned models or optimization steps, and can be computed with linear or near-linear time complexity.

We perform a large-scale empirical study comparing these simple methods to both the original OT formulation and recent temporal extensions. Across 32 benchmarks and three downstream RL algorithms, along with both online and offline settings, we find that our simple methods consistently match or exceed the performance of OT-based approaches. In particular, the Segment-matching method achieves the best results in several environments, and the Minimum-Distance method performs surprisingly well despite its complete lack of temporal modeling.

We make the following contributions:

1. Our results show that temporal alignment provides greater benefits to proximity-based IRL than relying on more elaborate proximity approximations. In particular, lightweight, temporally-aligned segment-matching rewards—constructed without solving the optimal transport coupling—consistently outperform OT across diverse RL settings. Moreover, as established in Theorem 1, the sufficiency of the simple Min-Dist approach highlights theoretical regimes where OT optimization offers no clear advantage.

2. Our proposed algorithms are conceptually simple, computationally efficient, and have strong empirically performance. At the same time, they offer insights into improving existing IRL methods without introducing unnecessary complexity.

3. With extensive experiments and a variety of settings, we compare the performance of our algorithms with serval popular and effective OT-based methods in the literature(Haldar et al., 2022; 2023), Moreover, we are the first to run a thorough ablation study on IRL+ReBRAC for this line of research.

## 2 PRELIMINARY

A standard episodic finite-horizon Markov Decision Process (MDP) can be represented as a tuple:

$$\mathcal{M} = (\mathcal{S}, \mathcal{A}, R, P, \rho_0, \gamma, T),$$

where $\mathcal{S}$ and $\mathcal{A}$ are the state and action spaces, $R : \mathcal{S} \times \mathcal{A} \to \mathbb{R}$ is the reward function, $P : \mathcal{S} \times \mathcal{A} \to \Delta(\mathcal{S})$ is the transition function, $\rho_0$ is the initial state distribution, $\gamma \in (0, 1]$ is the discount factor, and $T$ is the maximal episode length. The objective of an RL algorithm is to learn an optimal policy $\pi^*(a|s)$ that maximizes the expected cumulative discounted reward: $\pi^* = \arg\max_{\pi \in \Pi} \mathbb{E}_\pi \left[ \sum_{t=0}^{T-1} \gamma^t R(s_t, a_t) \right], s_0 \sim \rho_0, \ a_t \sim \pi(\cdot|s_t), \ s_{t+1} \sim P(\cdot|s_t, a_t)$ where $\Pi$ is the feasible policy set.

**Online and Offline RL** In online RL, the agent interacts with the environment during training to collect trajectories and update its policy. At each step, the agent selects an action from the current policy, receives a reward, and observes the next state. This interaction allows the agent to continuously improve its policy towards the RL objective via trial and error. However, online interactions can be inefficient, costly and risky, making online RL learning not always feasible. Offline RL then aims to learn purely from the pre-collected trajectory dataset labeled by task rewards, and determine a policy which can achieve the largest possible cumulative reward when it is deployed to interact with the MDP (Levine et al., 2020). The methodology that we develop in this paper applies to both online and offline setups, and we will be providing experimental results for both.

**Inverse RL** In many real-world scenarios, task-specific rewards are often inaccessible, nontrivial to define, or difficult to engineer. Different from offline RL problems, in Imitation Learning (IL), we learn from a dataset containing expert state-action pairs without reward signals. Specifically, in IL we have at our disposal a dataset $\mathcal{D}_e = \{\tau_i^e\}_{i=1}^K = \{(s_0^e, a_0^e, s_1^e, a_1^e, \dots s_{T_i}^e, a_{T_i}^e)_i\}_{i=1}^K$, which comprises $K$ expert demonstrations generated by some unknown expert policy $\pi_e$. Approaches to IL problems can mainly be classified into two categories: $(i)$ Behavior Cloning (BC) determines the policy by directly cloning the expert actions in the dataset; $(ii)$ Inverse RL (IRL) infers a reward function $R^*$

under which the underlying expert policy $\pi_e$ generating the dataset is optimal:

$$\mathbb{E}_{\pi_e}\left[\sum_{t=0}^{T-1} \gamma^t R^*(s_t, a_t)\right] \geq \mathbb{E}_{\pi}\left[\sum_{t=0}^{T-1} \gamma^t R^*(s_t, a_t)\right], \quad \forall \pi \in \Pi$$

After learning a reward $R^*$, we can employ standard RL algorithms to find a policy that mimics the expert. IRL avoids manual reward design while enabling the agent to acquire high-performing behavior through demonstration-driven learning. In this paper, we will be using the IRL approach for imitation learning.

## 3 IRL VIA OPTIMAL TRANSPORT

We will first describe the offline version of the problem. Following the setup in Luo et al. (2023), we assume we have access to a dataset $D$ consisting of both expert and non-expert trajectories. The trajectories do not include a reward signal, but we are informed which of the trajectories are expert trajectories.

### 3.1 THE OPTIMAL TRANSPORT REWARD ALGORITHM

The Optimal Transport Reward (OTR) heuristic first applies an optimal transport formulation to assign rewards to all trajectories in $D$, and subsequently uses an offline reinforcement learning algorithm to derive a high-performing policy (Luo et al., 2023). For concreteness, we assume that $D$ contains a single expert trajectory, denoted by $\tau^e = (s_1^e, a_1^e, \ldots, s_{T_e}^e, a_{T_e}^e)$, where $T_e$ is the length of the expert trajectory.

Let $\tau = (s_1, a_1, \ldots, s_T, a_T)$ denote a non-expert trajectory in $D$. Let $c(s_i, s_j^e)$ represent the transport cost (or dissimilarity) between a state $s_i$ in the non-expert trajectory and a state $s_j^e$ in the expert trajectory. This cost may be computed, for instance, as the Euclidean or cosine distance between the two states. The OTR algorithm computes a minimal-cost coupling between the non-expert and expert trajectories by solving the following Wasserstein distance objective:

$$\mathcal{W}(\tau, \tau^e) = \min_{\mu \in \mathbb{R}_+^{T \times T_e}} \sum_{i=1}^{T} \sum_{j=1}^{T_e} c(s_i, s_j^e)\mu_{i,j}, \quad \text{s.t.} \quad \mu\mathbf{1} = \frac{1}{T}\mathbf{1}, \quad \mu^\top\mathbf{1} = \frac{1}{T_e}\mathbf{1},$$

where $\mu$ is the coupling matrix that specifies the transport of probability mass between the two trajectories. Let $\mu^*$ denote the optimal transport plan that minimizes the above objective. The OTR algorithm then defines the reward for each state $s_i$ in the non-expert trajectory as

$$r(s_i) = -\sum_{j=1}^{T_e} c(s_i, s_j^e)\mu_{i,j}^*$$

Once rewards for all states in $D$ are assigned in this manner, an offline deep reinforcement learning algorithm is applied to learn the final policy.

### 3.2 TEMPORAL OPTIMAL TRANSPORT REWARD

The OTR approach has attracted significant attention in the literature (Fu et al., 2024; Wang et al., 2024a; Bobrin et al., 2024). However, the literature has also argued that the above OTR reward-labeling approach has two drawbacks (Fu et al., 2024). The first drawback is that the OT solution is *temporally invariant*. The Wasserstein distance treats trajectories as unordered sets of states, ignoring the sequence order. The consequence of such an omission entails that two non-expert trajectories differing in only temporal order will be assigned the same set of OT rewards. The second drawback is that OT-based rewards are inherently *non-stationary*, as the reward for a state depends on the entire trajectory, including temporally distant states. This global coupling can potentially introduce instability, because the resulting local reward for the same state transitions can differ.

**Context-Aware Cost Matrix** Addressing these two drawbacks, TemporalOT introduces two modifications that together can potentially improve the quality and fidelity of the IRL rewards (Fu et al., 2024). It first incorporates local temporal context by changing the pairwise cost matrix to a groupwise cost matrix. The context-aware cost $\tilde{c}(\cdot, \cdot)$ between a non-expert state $s_i \in \tau$ and an expert state $s_j^e \in \tau^e$ is defined as:

$$\tilde{c}(s_i, s_j^e) = \frac{1}{k_c} \sum_{h=0}^{k_c-1} c(s_{i+h}, s_{j+h}^e)$$

where $k_c$ is the context length; and $c(s_i, s_j^e)$ is the original pairwise cost between states. The resulting context-aware cost matrix smooths and improves proximity estimation by leveraging information from neighboring states.

**Temporal-masked OT Objective** To inject temporal alignment constraints into the OT objective, TemporalOT proposes to solve a Temporal-masked OT objective with the following matrix form (Gu et al., 2022; Fu et al., 2024):

$$\mu^* = \arg\min_{\mu \in \mathbb{R}_+^{T \times T}} \left\langle M \odot \mu, \tilde{C} \right\rangle_F - \epsilon \mathcal{H}(M \odot \mu)$$

$$\text{s.t.} \quad (M \odot \mu)\mathbf{1} = \frac{1}{T}\mathbf{1}, \ (M \odot \mu)^\top \mathbf{1} = \frac{1}{T}\mathbf{1}$$

where $\odot$ is the Hadamard product operator; $\langle \cdot, \cdot \rangle_F$ is the Frobenius inner product; $M$ is a mask matrix; $\tilde{C}_{i,j} = \tilde{c}(s_i, s_j^e)$ represents the context-aware cost matrix; $\mathcal{H}$ is the entropy regularizer for the masked coupling. Fu et al. (2024) selects a diagonal-like mask matrix with the width $k_m$ of the temporal alignment window: $M_{i,j} = 1$, if $j \in [i - k_m, i + k_m]$ and $M_{i,j} = 0$ otherwise. Such a temporal mask constrains the coupling to pairs of states that are only temporally close and reduces influence from states outside of the mask window. In practice, both the original OT and the new objective can be solved by the entropy-regularized Sinkhorn's algorithm (Cuturi, 2013).

While OT is a principled and structured approach to match trajectories, its direct application to reward labeling can lead to suboptimal imitation learning. In addition to TemporalOT, other recent methods have also made efforts to constrain or modify the original OT formulation by imposing some handcrafted temporal structure to improve the quality of OT-based rewards (Wang et al., 2024a; Bobrin et al., 2024).

## 4 METHODS: SIMPLE REWARD FUNCTIONS

Although the OT approaches for solving the IRL problem described in the previous section have been successfully applied to a variety of environments and datasets, all the approaches require understanding and solving a "sophisticated" optimal transport problem. In this work, we ask a fundamental question:

*Following the principle of Occam's razor, are there simpler, more direct IRL approaches which do not involve the intermediate step of solving the OT problem (or some other optimization problem)? And do these simpler approaches achieve similar, or even better, performance than the OT approaches?*

If the answers to these questions are "yes," then the findings would challenge the perceived necessity of using optimal transport for IRL, and advocate for a rethinking of design strategies for future IRL methods.

### 4.1 MINIMUM-DISTANCE REWARD

We start by proposing perhaps the most primitive and minimalist reward labeling method one can imagine, which we refer to as Minimum-Distance Reward (Min-Dist). It is simply the negative point-to-set distance between the non-expert state $s_t \in \tau$ and the expert demonstration $\tau^e$:

$$r_{\min}(s_t) := - \min_{s^e \in \tau^e} \text{Dist}(s_t, s^e)$$

where $\text{Dist}(\cdot, \cdot)$ is a suitable distance metric (e.g., Euclidean or cosine distance) between two states. The intuition is to treat any non-expert state close to the expert demonstration as a "good state" to learn from, without considering any temporal order.

## 4.2 Segment-matching Reward

We now propose a second reward function that takes into account the temporal alignment between the agent and the expert trajectory, but still does not involve solving an OT problem or any other optimization problem as an intermediate step. The Segment-matching Reward (Seg-match) segments the expert trajectory into $T$ consecutive non-overlapping pieces as evenly as possible. Recall that $T$ is the length of the non-expert trajectory for which we are labeling the rewards. Initially assume that $T \leq T_e$. More precisely, we partition the expert trajectory into $T$ contiguous segments $\{\Gamma_1, \Gamma_2, \ldots, \Gamma_T\}$, so that each state $s_t$ in the non-expert trajectory $\tau$ corresponds to one expert segment $\Gamma_t$. Let $q = \lfloor \frac{T_e}{T} \rfloor$ and $l = T_e \bmod T$. Each segment is thus defined by:

$$\Gamma_t = \{s_i^e \mid i \in [a_t, \ b_t] \cap \mathbb{Z}\},$$

where

$$a_t = (t-1)q + 1 + \min(t-1, l), \quad b_t = tq + \min(t, l)$$

Note that if $T_e = nT$ for some integer $n$, then $\Gamma_t = \{s_i^e \mid i \in [n(t-1)+1, nt] \cap \mathbb{Z}\}$, and if $T_e = T$, then $\Gamma_t = \{s_t^e\}$. In general, the first $l = T_e \bmod T$ segments receive one extra state to make sure $T$ segments fully partition the expert trajectory in the end.

For each $s_t \in \tau$, the reward is then taken as the negative minimum distance to states in its corresponding expert segment $\Gamma_t$:

$$r_{\text{seg}}(s_t) := - \min_{s^e \in \Gamma_t} \text{Dist}(s_t, s^e)$$

Note that if $T_e = T$, then we simply have $r_{\text{seg}}(s_t) = -\text{Dist}(s_t, s_t^e)$

Similar to Temporal OT and the literature of learning from demonstration (Fu et al., 2024; Liu et al., 2024), this heuristic exploits an implicit assumption that the non-expert has a similar movement speed to that of the expert agent. When $T \approx T_e$, the distance between a non-expert state and the corresponding expert segment is aligned with the temporal information. When $T \ll T_e$, unless the non-expert trajectory approaches the expert destination quickly, the distances will be large because non-expert states and expert states are not temporally consistent.

Note that the definition so far considers only the situation where $T \leq T_e$. It is also possible that the expert demonstration is shorter than the non-expert trajectory. In this case, for any timestep $t > T_e$, we simply compute the distance between $s_t \in \tau$ and the last expert state:

$$r_{\text{seg}}(s_t) = -\text{Dist}(s_t, s_{T_e}^e)$$

The intuition behind this is that for any state beyond the expert horizon, the quality of the non-expert state should be in negative proportion to its distance to the last known expert state, since no further demonstrations are available.

We have introduced four reward functions: simple and OT-based rewards. Let $d$ denote the dimension of the state and assume $T_e = T$. We summarize the computational complexity of the four methods in Table 1. Thus, not only are the Seg-Match and Min-Dist methods conceptually much simpler than OT approaches, but they also require less running time in practice. In Appendix C, we expand the discussion on their complexity and also present a more comprehensive simple reward formulation, which includes both Min-Dist and Seg-match as special cases.

Table 1: Computational complexity of different reward labeling methods.

| Algorithms | OT | TemporalOT | Seg-Match | Min-Dist |
|---|---|---|---|---|
| **Complexity** | $\mathcal{O}(dT^2)$ | $\mathcal{O}\left((k_c + d)T^2\right)$ | $\mathcal{O}(dT)$ | $\mathcal{O}(T \log T + dT)$ |

## 5 Experiments

In this section, we conduct a comprehensive evaluation of the following methods: the original OT rewards (OT) by Luo et al. (2023), the OT rewards with temporal alignment (TemporalOT) (Fu et al., 2024), and our two simple rewards, Seg-match and Min-Dist, proposed in Section 4. We do this for several downstream RL algorithms in order to explore the robustness of our findings. We further

include the baseline performance (referred to as the Oracle) of any downstream RL algorithms trained with the ground-truth reward function. And for the offline evaluation, the performance of vanilla behavior cloning is added as well. We use cosine distance to label rewards throughout the experiments in the main body. In Appendix A.2, we also include some experiments by using Euclidean distance.

In Section 5.1, we present the experimental results when applying IRL to the offline RL setting with two downstream offline RL algorithms. We then consider the online RL setting in Section 5.2. In Section 5.3, we study how different methods scale with more expert demonstrations. Lastly, Theorem 1 in Section 5.4 provides insight into our empirical results and explains why OT is unnecessary from a theoretical perspective.

## 5.1 OFFLINE RL EVALUATION

We evaluate all methods on the canonical D4RL benchmark (Fu et al., 2021), which consists of three domains: `MuJoCo-v2`, `Adroit-v1`, and `Antmaze-v2`. Each domain provides different environments with corresponding datasets of different quality. In total, there are 23 individual datasets to be tested on. Following the same experimental setup as Luo et al. (2023), for each D4RL dataset, the unlabeled offline dataset $\mathcal{D}_o$ is obtained by removing all ground-truth rewards, and the *one* expert trajectory with the highest ground-truth episodic return among all trajectories in $\mathcal{D}_o$. Then, we apply the reward labeling methods to generate proxy rewards for the offline dataset $\mathcal{D}_o$. Similar to what is done for the OT rewards, we apply reward post-processing, first with an exponential squashing function and then rescaling with the the global dataset statistics (details in Appendix A.1). After the reward relabeling, we are ready to use any offline RL algorithms to extract a policy.

**IRL + IQL**  We first consider the Implicit Q-Learning (IQL) (Kostrikov et al., 2021) algorithm as the downstream RL algorithm, which is also used by the original OTR method. We employ the same IQL code used by the official implementation of OTR (Luo et al., 2023). We use the IQL hyperparameters recommended by the OTR authors and keep *all of them* the same throughout the experiments, except for enabling state normalization for IQL training on `MuJoCo` and `Adroit` benchmarks, as we found that this common trick can improve performance in most cases, including for the IQL oracle.

Figure 1 and Table 2 report the D4RL normalized scores with standard deviations for the five reward functions for a total of 23 datasets. For each experiment, we run 10 random seeds. Instead of reporting the performance from the very last evaluation during the training process, we report the average performance over the last four evaluations to offset result fluctuations over timesteps (Wang et al., 2024b). A score is highlighted if it is larger than 95% of the highest scores in that row, but the oracle and BC[1] performances are not highlighted for comparison. Appendix A.2 provides more details about the offline experimental setup.

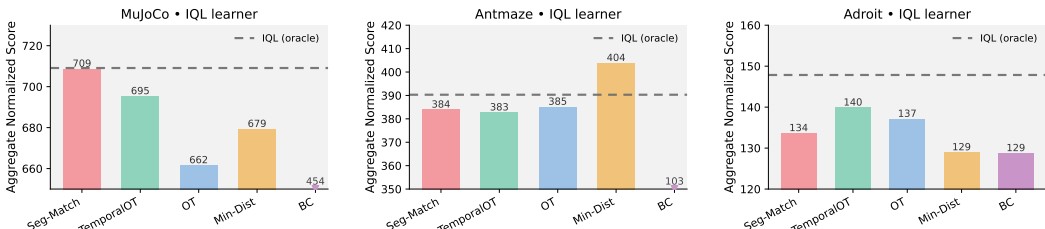

Figure 1: Aggregate normalized scores with IQL as the downstream RL algorithm.

Figure 1 and Table 2 show that *the Segment-matching approach provides the overall best performance*. Surprisingly, *the exceedingly simple Min-Dist approach also does quite well*. It can obtain the best total score on the `Antmaze` domain and also surpass the original OT approach on the `MuJoCo` domain. In addition, the superior performance of TemporalOT and Seg-match approaches on the `MuJoCo` domain verifies the importance of temporal alignment, which is missing from the original OT approach. Although Seg-Match does not achieve the best compared with OT, we emphasize that large variances are observed in `Adroit` (see Table 2), and thus their performance is on par with each other.

---

[1]BC performances are copied from Table 1 provided by Tarasov et al. (2023b)

More recent IRL work, e.g., CLUE (Liu et al., 2023) claims SOTA performance by also taking a proximity-based approach. However, they add extra complexity to train state representations and then approximate distance to expert demonstrations in the learned embedding space. Building such a state feature extractor is incompatible with the online RL setting, where no pre-collected trajectories are available. Furthermore, their methods are sensitive to the choice of hyperparameters. Even with hyper-parameter tuning, CLUE does not show a clear advantage over Seg-Match on the MuJoCo benchmark. In Appendix A.5, we include the performance of related methods in the literature.

**IRL + ReBRAC**    Related work often argues that their IRL methods can be combined with arbitrary offline RL, but only applies IQL for evaluation. We are the *first* work to extensively test with the ReBRAC downstream learner, a SOTA behavior-regularized actor-critic method, which regularizes the online TD3 algorithm in a minimalist manner (Tarasov et al., 2023a). The IQL and ReBRAC algorithms take two very different approaches to offline DRL. Therefore, we would like to know whether the different reward labeling approaches are robust to the design of offline RL algorithms.

We employ the ReBRAC implementation from the Clean Offline RL (CORL) codebase (Tarasov et al., 2023b). The general experimental setup and evaluation criterion are the same as IQL. In our experiments, we found that both OT-based and simple rewards are sensitive to the choice of actor and critic regularization coefficients required in the ReBRAC algorithm[2]. To demonstrate the full capability of each method, substantial tuning for these two hyperparameters is conducted to obtain the best performance for `MuJoCo` datasets. We further describe this issue and provide details for tuning hyperparameters in Appendix A.3.

Figure 2 and Table 6 summarize the normalized score for the nine `MuJoCo` datasets. The ReBRAC oracle performance is copied from the original paper (Tarasov et al., 2023a). Surprisingly once again, the two simple reward functions provide the same total performance as two OT-based approaches. In Appendix A.3, we also provide the tuned results of OT and Seg-match for `Adroit-pen` and `Antmaze` datasets. Again, Seg-match shows a total performance on par with the OT reward. Note that, even in the case we adopt the default ReBRAC hyperparameters, Min-Dist exhibits the best total performance among the four IRL approaches as shown in Table 5. Therefore, these results further question the need for OT's complex design (involving an optimization problem).

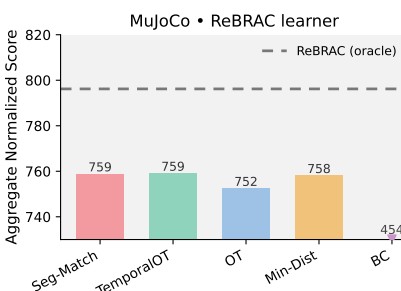

Figure 2: Aggregate normalized scores with ReBRAC as the downstream RL algorithm.

## 5.2 ONLINE RL EVALUATION

We then evaluate our proposed reward relabeling methods in the online reinforcement learning setting. As in Fu et al. (2024), we assume that: ($i$) two expert trajectories are available; ($ii$) no non-expert trajectories are available; ($iii$) the agent interacts with the environment during training to collect reward-less state and action trajectory data. This setup tests whether rewards estimated from limited expert demonstrations can effectively guide long-horizon exploration and policy optimization without access to the underlying task rewards. Experiments are conducted on the MetaWorld benchmark (Yu et al., 2020).

For policy learning, we adopt DrQ-v2 (Kostrikov et al., 2022), an off-policy actor-critic algorithm that combines sample-efficient Q-learning with strong visual data augmentation. Policies are trained from scratch using only the rewards generated by the IRL methods, without access to the environment's ground-truth reward. Note that, unlike offline RL evaluations, where we can compute global dataset statistics to normalize the relabeled rewards, online RL maintains a dynamic replay buffer and thus uses a different strategy for reward post-processing. More details about task configurations, implementation details, and training hyperparameters can be found in Appendix B.

---

[2]With critic regularization being 0 and actor regularization being 1, ReBRAC can be algorithmically regarded as the TD3+BC algorithm (Fujimoto & Gu, 2021), so we also test TD3+BC as the downstream RL algorithm in Appendix A.3.

In the MetaWorld benchmark, all task episodes have a fixed length, which reduces the Segment-matching method to one-to-one matching between identical agent and expert timesteps. While this preserves temporal order, it is less aware of the local trajectory structure and risks of large reward variance along an episode. A simple remedy is to smooth the relabeled episodic rewards by a Gaussian kernel with a standard deviation of 0.5.

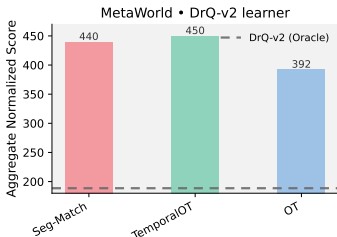

Figure 3: Aggregate normalized scores for online RL on Meta-World with DrQ-v2 as the downstream RL algorithm. DrQ-v2 (Oracle) denotes performance obtained from training with ground-truth **sparse** task reward.

In Figure 3 and Table 12, we show the success rate (multiplied by 100) of each task. We repeat each experiment on 5 random seeds. Similarly, the average success rate over the last four evaluations is reported. The DrQ-v2 oracle results are borrowed from Fu et al. (2024), and we omit the Min-Dist method as it fails in these challenging online tasks.

The results reveal several notable trends. The Segment-matching method performs competitively with TemporalOT and greatly outperforms the standard OT baseline, despite its simplicity, suggesting that simple proximity is enough for reward labeling. In aggregate, both TemporalOT and Segment-window substantially outperform the standard OT baseline, highlighting the importance of temporal structure in reward design. These findings align with our offline results and reinforce one of the central themes of this work: *simple proximity-based heuristics, when combined with simple temporal inductive biases, can match or exceed the performance of more complex OT-based methods in both offline and online reinforcement learning settings.*

### 5.3 MULTI-EXPERT GENERALIZATION

We study how performance scales with the number of expert demonstrations $K \in \{1, 5, 10, 20\}$ in the offline setting, using IQL as the downstream learner. For $K > 1$, following OT and TemporalOT, we compute proximity rewards with respect to *each* expert trajectory and retain the set of rewards from the expert that yields the highest relabeled episode return. Figure 4 reports the normalized score aggregated across all offline benchmarks. We run 5 seeds for this set of experiments.

Figure 4 highlights four observations. **(i)** Temporal alignment improves performance across $K$ values. The performance curves of Seg-Match and TemporalOT are consistently above those of Min-Dist and OT. **(ii)** Temporal alignment maintains performance with increasing $K$. Methods lacking the temporal structure, Min-Dist and OT, exhibit declining performance as $K$ increases. In contrast, Seg-Match and TemporalOT can be improved with more expert demonstrations. **(iii)** Sophisticated proximity approximation fails to yield a clear advantage. Seg-Match is the best or at least competitive at most $K$.

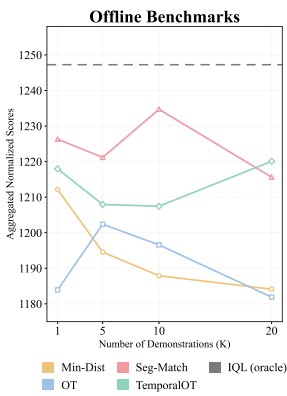

Figure 4: Aggregated D4RL performance as a function of the number of experts ($K$).

This scaling study once again enhances the central claim that temporal alignment plays a more crucial role than proximity approximation for trajectory-matching reward design. In Appendix A.4, we include similar plots for 3 offline domains separately.

### 5.4 MIN-DISTANCE IS SUFFICIENT FOR TRAJECTORY MATCHING

In this section, we provide insights on why the simple `min` operator is enough for proximity approximation. Inspired by Yu et al. (2022), Theorem 1 gives a bound on OT's ability to follow the

expert trajectory in the offline RL setting. The complete problem formulation and proof can be found in Appendix D.

**Theorem 1** (Sufficiency Bound; condensed version). *Under mild regularity conditions on concentration of relabeled OT rewards $r_{s,a}^{ot}$, with high probability $\geq 1 - \delta$, the optimal trajectory-matching policy learned by OT performs no better than Min-Dist rewards by the following bound:*

$$J(\pi_{ot}^*) - J(\pi_{min}^*) \leq \frac{1}{1-\gamma}\mathbb{E}_{d^{\pi_\beta}}\left[\rho(s,a) \cdot \left(\mathcal{R}(s,a) - r_{s,a}^{min}\right)\right]$$

$$- \frac{1}{1-\gamma}\mathbb{E}_{d^{\pi_{min}^*}}\left[\rho(s,a) \cdot \left(\mathcal{R}(s,a) - r_{s,a}^{min}\right)\right]$$

$$- \frac{\alpha}{1-\gamma}\text{Div}(\pi_{min}^*, \pi_\beta) + \mathcal{O}(\frac{1}{1-\gamma})$$

*where $J(\pi)$ is the expected discounted return of $\pi$ in MDP; and $d^\pi(s,a)$ is the marginal state-action distribution of $\pi$; $\rho \in [0,1]$ estimates the percentage of mismatch between OT and Min-Dist relabeled rewards in the dataset; $\mathbb{R}$ is the average stationary OT reward function; $\text{Div}$ is a divergence metric.*

We point out several scenarios where this bound is small in Appendix D.

## 6 RELATED WORK

Inverse reinforcement learning (IRL) aims to infer reward functions from expert demonstrations, under which the expert behavior is optimal (Ng & Russell, 2000; Arora & Doshi, 2020). Recent advances have witnessed significant interest in leveraging Optimal Transport (OT) for IRL. OT offers a principled method for aligning expert and agent trajectories by solving for a minimum-cost coupling, typically formulated through the Wasserstein distance. Xiao et al. (2019) links adversarial imitation learning with OT theory by minimizing the Wasserstein distance from its dual formulation as smooth, meaningful reward functions. Later on, Dadashi et al. (2021) proposes to minimize the Wasserstein distance via its primal formulation instead, avoiding the potential optimization issues in the dual formulation. Motivated by OT's lack of temporal consistency, Wang et al. (2024a) builds on top of Soft Dynamic Time Warping (SDTW) to enforce temporal constraints such that two trajectories must be strictly aligned at the beginning and the end, and their alignment indices must be monotonically non-decreasing in time. In addition to the efforts devoted to better constraining the OT formulation, Bobrin et al. (2024) emphasizes aligning state representations in a latent space learned by Intention Conditioned Value Function (ICVF) instead of the original state space. For better alignment, only the tail of the expert's trajectory is considered, meaning that alignment starts at the nearest first expert state to the agent's starting position according to the cost matrix.

OT-based trajectory-matching has other broader applications. For example, it has been used for online fine-tuning of offline robotic policy in real-world tasks (Haldar et al., 2022; 2023). It can also be extended to align trajectories from different state spaces (domains), and thus solves the cross-domain imitation learning problem (Fickinger et al., 2022; Lyu et al., 2025). Recently, Asadulaev et al. (2024) even expands OT usage by framing the entire offline RL challenge as an optimal transport problem and proposes an algorithm that dynamically stitches trajectories from multiple behavior policies, highlighting OT's versatility beyond simple reward labeling.

## 7 CONCLUSION

Re-examining whether optimal transport (OT) is necessary for IRL, we find that the proposed simple approaches, Segment-matching and Min-distance, often match or beat OT across various RL settings, suggesting OT's perceived gains barely stem from solving for optimal couplings. Despite our empirical and theoretical analysis of the simple rewards, TemporalOT can sometimes perform slightly better, e.g., on `Adroit` and `Metaworld`. This indicates that the lightweight temporal structure injected in Segment-matching could be further improved to boost performance. Looking ahead, promising directions include aligning in learned representation spaces, favoring simpler but stronger temporal inductive biases over full OT, and stress-testing robustness under the evaluation protocol used in our work.

ETHICS STATEMENT

This work studies reward relabeling for imitation learning and offline/online RL. All experiments use public benchmarks (D4RL and MetaWorld) with synthetic or simulated environments; no human-subject data, personally identifiable information, or sensitive attributes were collected. Potential risks include (i) misuse of learned policies in real-world robotics without adequate safety checks; (ii) unintended biases carried over from benchmark datasets; and (iii) environmental costs of computation. To mitigate these risks, we (a) restrict evaluations to standard simulators; (b) report aggregate results across many seeds and tasks to reduce overclaiming; (c) release code and configuration files to enable scrutiny and replication; and (d) discuss limitations (e.g., reliance on proximity metrics and temporal assumptions) and the need for safety evaluation before any real-world deployment.

REPRODUCIBILITY STATEMENT

We make reproducibility a primary goal. Concretely:

- **Experimental setup:** Section 5 details benchmarks, downstream algorithms, and evaluation protocols ("avg4" vs. "avg1"). Offline RL uses **IQL** and **ReBRAC**; online RL uses **DrQ-v2**. Dataset sources and versions are specified in Section 5.1 (D4RL: MuJoCo-v2, Adroit-v1, Antmaze-v2) and Section 5.2 (MetaWorld).

- **Hyperparameters & implementation detail:** Post-processing of rewards, mask-ing/windowing, distance metrics, and learner hyperparameters are documented in the text and in the appendix (see App. A.1, A.2, B, A.3).

- **Randomness and compute:** We report the number of seeds for each setting (typically *10* for offline IQL tables/curves and *5* for MetaWorld). Seeds are fixed and logged. Hardware requirements are modest (single-GPU training per run); exact commands and run scripts are provided.

- **Complete results:** Domain totals and per-task tables are included in the main text (e.g., Tables 2, 6) and appendix; ablations (multi-expert $K$) are shown in Fig. 4 and App. A.4.

We will release all code, figure generation scripts as supplementary material and via repository, enabling reproduction of tables and plots.

USE OF LARGE LANGUAGE MODELS (LLMS)

We used an LLM-based assistant *only* for language polishing (grammar, phrasing, and layout of figure captions/section headers). All technical content, including ideas, algorithms, experiments, analyses and conclusions, was conceived, implemented, and verified by the authors. The model was not used to generate or modify experimental results, proofs, or related-work claims. Authors carefully reviewed and edited all text for factual accuracy and proper citation. No confidential or proprietary data were provided to the LLM.

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

## A  OFFLINE RL EVALUATIONS

### A.1  REWARD POST-PROCESSING

To prevent instability during policy optimization, especially for value-based RL algorithms, we first apply a standard exponential squashing function to all rewards produced by IRL. This transformation helps mitigate the implicit penalization of longer trajectories and controls the variance of reward magnitudes. For any assigned raw reward $r_t$, we apply:

$$r_t \leftarrow \alpha \cdot \exp(\beta \cdot r_t)$$

where $\alpha > 0$ and $\beta > 0$ are scaling and sharpness parameters. Following Dadashi et al. (2021); Luo et al. (2023), we use fixed values $\alpha = 5$, $\beta = 5$ throughout all experiments for OT and TemporalOT rewards. For our proposed simple rewards, we want to keep them simple and have as few hyperparameters as possible. We instead use $\alpha = 1$, $\beta = 1$, as prior work suggests that performance is generally robust to these hyperparameters.

The second reward post-processing happens after exponential squashing. As is commonly done for offline RL algorithms (Kostrikov et al., 2021; Fujimoto & Gu, 2021; Luo et al., 2023), we apply reward rescaling and reward bias for the squashed rewards:

$$r_t \leftarrow \texttt{reward\_scale} \cdot r_t + \texttt{reward\_bias}$$

Here, the rescaling factor is $\frac{1000}{\texttt{max\_return - min\_return}}$, where $\texttt{max\_return}$ is the largest relabeled return; $\texttt{min\_return}$ is the smallest relabeled return among all offline trajectories. We adopt the same reward offset as Luo et al. (2023). For $\texttt{MuJoCo}$ and $\texttt{Adroit}$, it is 0, while $\texttt{Antmaze}$ uses -2.

This post-processing step is used consistently across all OT-based and proximity-based reward functions evaluated in our offline RL study.

### A.2  IQL EXPERIMENTS

Table 2: Normalized scores with IQL as downstream RL algorithm.

| Dataset | BC | IQL (oracle) | OT | TemporalOT | Seg-match | Min-Dist |
|---|---|---|---|---|---|---|
| hopper-medium-replay | $29.81 \pm 2.07$ | $94.66 \pm 8.2$ | $65.59 \pm 20.5$ | $77.01 \pm 4.9$ | $\mathbf{87.69} \pm 1.1$ | $\mathbf{86.96} \pm 2.1$ |
| hopper-medium | $53.51 \pm 1.76$ | $63.50 \pm 4.8$ | $\mathbf{73.68} \pm 4.8$ | $\mathbf{76.30} \pm 4.9$ | $76.77 \pm 4.5$ | $72.39 \pm 4.2$ |
| hopper-medium-expert | $52.30 \pm 4.01$ | $103.67 \pm 8.9$ | $\mathbf{105.45} \pm 7.6$ | $104.51 \pm 11.5$ | $\mathbf{107.85} \pm 4.0$ | $82.67 \pm 37.7$ |
| halfcheetah-medium-replay | $35.66 \pm 2.33$ | $43.12 \pm 1.6$ | $\mathbf{41.53} \pm 0.4$ | $43.15 \pm 0.6$ | $41.03 \pm 1.0$ | $\mathbf{42.38} \pm 0.9$ |
| halfcheetah-medium | $42.40 \pm 0.19$ | $47.49 \pm 0.3$ | $43.46 \pm 0.3$ | $\mathbf{45.13} \pm 0.2$ | $43.23 \pm 0.3$ | $\mathbf{45.03} \pm 0.2$ |
| halfcheetah-medium-expert | $55.95 \pm 7.35$ | $91.24 \pm 2.3$ | $\mathbf{88.53} \pm 3.4$ | $89.49 \pm 3.5$ | $\mathbf{90.71} \pm 2.7$ | $88.72 \pm 2.9$ |
| walker2d-medium-replay | $21.80 \pm 10.15$ | $73.79 \pm 8.9$ | $53.87 \pm 19.5$ | $\mathbf{69.73} \pm 11.4$ | $\mathbf{70.93} \pm 14.6$ | $70.82 \pm 8.6$ |
| walker2d-medium | $63.23 \pm 16.24$ | $80.57 \pm 4.2$ | $\mathbf{79.58} \pm 1.6$ | $79.83 \pm 1.8$ | $80.20 \pm 2.4$ | $\mathbf{79.43} \pm 1.1$ |
| walker2d-medium-expert | $98.96 \pm 15.98$ | $111.06 \pm 0.2$ | $109.93 \pm 0.2$ | $\mathbf{110.19} \pm 0.2$ | $110.31 \pm 0.2$ | $\mathbf{110.91} \pm 0.2$ |
| **MuJoCo-total** | 453.6 | 709.10 | 661.62 | **695.34** | **708.73** | 679.33 |
| pen-human | $71.03 \pm 6.26$ | $69.86 \pm 17.2$ | $68.20 \pm 17.3$ | $\mathbf{71.43} \pm 20.8$ | $\mathbf{72.28} \pm 17.8$ | $66.73 \pm 19.1$ |
| pen-cloned | $51.92 \pm 15.15$ | $70.96 \pm 20.3$ | $\mathbf{61.50} \pm 18.8$ | $\mathbf{62.13} \pm 19.5$ | $54.78 \pm 17.5$ | $55.97 \pm 20.1$ |
| door-human | $2.34 \pm 4.00$ | $3.34 \pm 1.6$ | $\mathbf{3.26} \pm 1.8$ | $3.07 \pm 1.7$ | $3.10 \pm 1.8$ | $2.87 \pm 1.5$ |
| door-cloned | $-0.09 \pm 0.03$ | $0.45 \pm 1.0$ | $\mathbf{1.28} \pm 1.4$ | $0.43 \pm 0.7$ | $0.46 \pm 1.0$ | $0.62 \pm 1.0$ |
| hammer-human | $3.03 \pm 3.39$ | $1.78 \pm 0.4$ | $1.82 \pm 0.4$ | $\mathbf{1.97} \pm 0.8$ | $1.88 \pm 1.3$ | $1.77 \pm 0.5$ |
| hammer-cloned | $0.55 \pm 0.16$ | $1.30 \pm 0.7$ | $\mathbf{0.92} \pm 0.7$ | $0.79 \pm 0.3$ | $\mathbf{0.92} \pm 0.7$ | $0.87 \pm 0.5$ |
| relocate-human | $0.04 \pm 0.03$ | $0.19 \pm 0.3$ | $0.14 \pm 0.2$ | $0.09 \pm 0.1$ | $\mathbf{0.20} \pm 0.3$ | $0.10 \pm 0.1$ |
| relocate-cloned | $-0.06 \pm 0.01$ | $-0.03 \pm 0.1$ | $0.01 \pm 0.1$ | $-0.02 \pm 0.1$ | $0.03 \pm 0.1$ | $\mathbf{0.04} \pm 0.1$ |
| **Adroit-total** | 128.76 | 147.85 | **137.14** | 139.89 | 133.65 | 128.97 |
| antmaze-umaze-play | $55.25 \pm 4.15$ | $86.35 \pm 3.5$ | $81.30 \pm 5.4$ | $82.05 \pm 4.0$ | $78.10 \pm 5.3$ | $\mathbf{88.98} \pm 2.8$ |
| antmaze-umaze-diverse | $47.25 \pm 4.09$ | $64.05 \pm 8.1$ | $66.45 \pm 6.6$ | $60.62 \pm 8.1$ | $68.00 \pm 11.7$ | $\mathbf{71.60} \pm 7.8$ |
| antmaze-medium-play | $0.00 \pm 0.00$ | $75.45 \pm 4.7$ | $\mathbf{75.03} \pm 4.8$ | $72.75 \pm 4.6$ | $69.83 \pm 4.3$ | $73.50 \pm 4.8$ |
| antmaze-medium-diverse | $0.75 \pm 0.83$ | $72.92 \pm 5.4$ | $72.53 \pm 5.3$ | $\mathbf{73.05} \pm 3.7$ | $71.80 \pm 5.2$ | $70.98 \pm 4.0$ |
| antmaze-large-play | $0.00 \pm 0.00$ | $46.38 \pm 5.5$ | $44.93 \pm 5.2$ | $45.67 \pm 6.5$ | $\mathbf{49.58} \pm 6.1$ | $48.03 \pm 6.2$ |
| antmaze-large-diverse | $0.00 \pm 0.00$ | $45.15 \pm 5.4$ | $44.92 \pm 5.0$ | $\mathbf{48.58} \pm 5.4$ | $46.62 \pm 7.0$ | $\mathbf{50.78} \pm 6.3$ |
| **Antmaze-total** | 103.26 | 390.30 | **385.15** | 382.73 | **383.93** | **403.85** |

We use the official implementation of OT reward (Luo et al., 2023) and thus adopt all the hyperparameters. Table 3 summarizes them and also includes the information about reward labeling and computation resources for each individual experiment.

Table 3: Hyperparameters for IQL offline evaluations.

| | Hyperparameter | Value |
|---|---|---|
| Training | Total training steps | 1e6 |
| | Evaluation frequency | 1e4, MuJoCo, Adroit |
| | | 5e4, Antmaze |
| | Evaluation episodes | 10, MuJoCo, Adroit |
| | | 100, Antmaze |
| | Batch size | 256 |
| | Seeds | `range(10)` |
| Network Architecture | Hidden layers | (256, 256) |
| | Dropout | None |
| | Network initialization | orthogonal |
| IQL | Discount factor | 0.99 |
| | Optimizer | Adam |
| | Policy learning rate | $3 \times 10^{-4}$, cosine decay to 0 |
| | Critic learning rate | $3 \times 10^{-4}$ |
| | Value learning rate | $3 \times 10^{-4}$ |
| | Target network update rate | $5 \times 10^{-3}$ |
| | Temperature | 3.0, MuJoCo |
| | | 0.5, Adroit |
| | | 10, Antmaze |
| | Expectile | 0.7, MuJoCo, Adroit |
| | | 0.9, Antmaze |
| | State normalization | True, Mujoco, Adroit |
| | | False, Antmaze |
| Reward Labeling | Episode length $T$ | 1000 |
| | Cost function | cosine distance |
| | Squashing function | $5.0 \cdot \exp(5.0 \cdot T \cdot r/|\mathcal{S}|)$, OT & TemporalOT |
| | | $\exp(r)$, Seg-match & Min-Dist |
| | Number of experts | 1 |
| Computation | Compute resources | Intel(R) Xeon(R) Platinum 8268 CPU |
| | Number of CPU workers | 1 |
| | Requested compute memory | 8 GB |
| | Approximate average execution time | 6 hrs |

**Evaluations with Euclidean Distance**   Throughout the main body, we adopt the commonly used cosine distance as the similarity metric for any two states when IRL methods compute the pseudo-rewards. Here, we also include evaluations by using Euclidean distance for IRL reward labeling in Table 4. Note that compared with using cosine distance, the performance in general declines, which indicates that Euclidean distance may struggle to embed and capture the geometric property, e.g., proximity or similarity, among states. Since none of the methods work for `Adroit`'s Door, Hammer, and Relocate tasks, we omit those evaluations.

Table 4: Normalized scores with IQL as downstream RL algorithm. Euclidean distance is used instead of cosine distance.

| Dataset | IQL (oracle) | OT | TemporalOT | Seg-match | Min-Dist |
|---|---|---|---|---|---|
| hopper-medium-replay | $94.67 \pm 8.2$ | $\mathbf{95.03} \pm 1.3$ | $\mathbf{96.05} \pm 1.0$ | $\mathbf{97.10} \pm 1.0$ | $\mathbf{96.17} \pm 1.3$ |
| hopper-medium | $63.51 \pm 4.7$ | $\mathbf{75.16} \pm 4.5$ | $73.78 \pm 3.8$ | $\mathbf{77.52} \pm 5.0$ | $71.59 \pm 3.7$ |
| hopper-medium-expert | $103.60 \pm 9.0$ | $\mathbf{108.12} \pm 3.2$ | $\mathbf{107.58} \pm 4.6$ | $\mathbf{109.14} \pm 2.7$ | $103.09 \pm 20.1$ |
| halfcheetah-medium-replay | $43.13 \pm 1.6$ | $39.01 \pm 2.0$ | $37.07 \pm 2.4$ | $38.46 \pm 2.6$ | $\mathbf{42.75} \pm 1.1$ |
| halfcheetah-medium | $47.49 \pm 0.3$ | $\mathbf{43.28} \pm 0.7$ | $42.56 \pm 0.5$ | $42.76 \pm 0.3$ | $\mathbf{45.11} \pm 0.6$ |
| halfcheetah-medium-expert | $91.06 \pm 2.2$ | $\mathbf{92.08} \pm 1.3$ | $58.51 \pm 6.7$ | $\mathbf{91.44} \pm 1.5$ | $\mathbf{92.56} \pm 1.4$ |
| walker2d-medium-replay | $73.79 \pm 8.9$ | $\mathbf{55.09} \pm 22.0$ | $\mathbf{56.72} \pm 14.7$ | $52.16 \pm 16.2$ | $49.97 \pm 24.4$ |
| walker2d-medium | $80.66 \pm 4.0$ | $\mathbf{79.65} \pm 0.7$ | $77.78 \pm 3.4$ | $72.50 \pm 4.8$ | $\mathbf{80.17} \pm 0.5$ |
| walker2d-medium-expert | $111.06 \pm 0.2$ | $\mathbf{110.76} \pm 0.1$ | $\mathbf{110.14} \pm 0.1$ | $109.47 \pm 0.2$ | $\mathbf{110.85} \pm 0.2$ |
| **MuJoCo-total** | 708.96 | **698.18** | 660.19 | **690.54** | **692.27** |
| pen-human | $68.78 \pm 18.1$ | $\mathbf{67.69} \pm 17.0$ | $\mathbf{69.67} \pm 22.5$ | $\mathbf{69.60} \pm 19.3$ | $\mathbf{69.40} \pm 21.1$ |
| pen-cloned | $70.62 \pm 20.0$ | $58.31 \pm 21.4$ | $57.44 \pm 18.3$ | $\mathbf{63.71} \pm 19.4$ | $\mathbf{61.02} \pm 20.0$ |
| **Adroit-total** | 139.40 | 126.00 | **127.12** | **133.32** | **130.42** |
| antmaze-umaze-play | $86.35 \pm 3.5$ | $89.03 \pm 2.9$ | $\mathbf{90.48} \pm 2.6$ | $\mathbf{91.80} \pm 2.9$ | $\mathbf{93.83} \pm 1.6$ |
| antmaze-umaze-diverse | $64.05 \pm 8.1$ | $62.95 \pm 12.2$ | $63.92 \pm 8.3$ | $\mathbf{69.00} \pm 7.6$ | $\mathbf{70.75} \pm 5.7$ |
| antmaze-medium-play | $75.45 \pm 4.7$ | $67.55 \pm 5.1$ | $\mathbf{70.72} \pm 6.2$ | $\mathbf{73.28} \pm 4.0$ | $63.03 \pm 5.4$ |
| antmaze-medium-diverse | $72.92 \pm 5.4$ | $\mathbf{67.17} \pm 5.1$ | $\mathbf{69.72} \pm 4.8$ | $\mathbf{69.12} \pm 4.8$ | $\mathbf{68.15} \pm 4.7$ |
| antmaze-large-play | $46.38 \pm 5.5$ | $\mathbf{49.17} \pm 5.7$ | $\mathbf{48.38} \pm 6.3$ | $42.90 \pm 6.3$ | $46.63 \pm 4.7$ |
| antmaze-large-diverse | $45.15 \pm 5.4$ | $\mathbf{50.97} \pm 4.9$ | $48.23 \pm 5.9$ | $45.75 \pm 5.7$ | $45.55 \pm 6.3$ |
| **Antmaze-total** | 390.30 | **386.85** | 391.45 | **391.85** | 387.93 |

### A.3 ReBRAC Experiments

**Hyperparameter Settings**  Because ReBRAC tunes hyperparameters for each individual dataset in D4RL, we refer readers to the official ReBRAC implementation[3] to learn all the details. We keep the default ReBRAC hyperparameters unchanged for the `MuJoCo` experiments in Table 5, but tune the BC regularization of both critic and actor for experiments in Tabel 6 and Table 7. The setup for ReBRAC reward labeling and computation requirements are the same as IQL experiments in Table 3.

**Hyperparameter Tuning for ReBRAC**  ReBRAC substantially tuned hyperparameters. e.g., critic BC regularization strength, actor BC regularization strength, critic/action learning rate, and batch size. As a result, each D4RL dataset inherits a set of optimal hyperparameters. However, such tuned hyperparameters are only optimal when training ReBRAC with ground-truth task rewards provided by the datasets. For example, D4RL `Antmaze` domain provides a sparse reward signal of 1 when the ant agent successfully reaches the goal before the environment timeout; otherwise, 0. To overcome the issue of reward sparsity, ReBRAC uses a larger discount factor $\gamma = 0.999$ to propagate the reward signal through TD learning faster. However, such a large discount factor is not suitable for dense reward tasks, like `MuJoCo`. So `MuJoCo` and `Adroit` use $\gamma = 0.99$. For the four IRL rewards discussed throughout this paper, they assign dense rewards to task episodes, including `Antmaze` domain. Therefore, $\gamma = 0.999$ is incompatible for the four reward labeling methods and indeed all of them fail in this case. In addition to the influence of the discount factor, performance is also sensitive to the BC regularizer strengths for both critic and actor training. This explains the abrupt failure of Segment-matching reward on `halfcheetah-medium` dataset in Table 5 if we adopt the default untuned regularizations.

Table 5: Normalized scores with ReBRAC as the downstream RL algorithm. The default **untuned** ReBRAC hyperparameters are used in this table.

| Dataset | BC | ReBRAC (oracle) | OT | TemporalOT | Seg-match | Min-Dist |
|---|---|---|---|---|---|---|
| hopper-medium-replay | $29.81 \pm 2.07$ | $98.1 \pm 5.3$ | $\mathbf{95.60} \pm 2.4$ | $96.35 \pm 2.5$ | $90.38 \pm 4.0$ | $\mathbf{95.99} \pm 2.3$ |
| hopper-medium | $53.51 \pm 1.76$ | $102.0 \pm 1.0$ | $95.00 \pm 6.8$ | $\mathbf{96.01} \pm 4.8$ | $73.68 \pm 30.7$ | $\mathbf{97.36} \pm 2.8$ |
| hopper-medium-expert | $52.30 \pm 4.01$ | $107.0 \pm 6.4$ | $104.75 \pm 6.2$ | $99.48 \pm 6.5$ | $\mathbf{108.17} \pm 2.2$ | $\mathbf{109.33} \pm 1.2$ |
| halfcheetah-medium-replay | $35.66 \pm 2.33$ | $51.0 \pm 0.8$ | $40.06 \pm 1.3$ | $42.67 \pm 0.9$ | $\mathbf{45.42} \pm 1.5$ | $43.54 \pm 0.4$ |
| halfcheetah-medium | $42.40 \pm 0.19$ | $65.6 \pm 1.0$ | $47.01 \pm 0.3$ | $\mathbf{49.69} \pm 0.5$ | $2.36 \pm 1.1$ | $\mathbf{48.92} \pm 0.5$ |
| halfcheetah-medium-expert | $55.95 \pm 7.35$ | $101.1 \pm 5.2$ | $\mathbf{92.27} \pm 1.1$ | $92.78 \pm 1.6$ | $89.99 \pm 2.6$ | $92.66 \pm 1.7$ |
| walker2d-medium-replay | $21.80 \pm 10.15$ | $77.3 \pm 7.9$ | $68.25 \pm 5.7$ | $67.80 \pm 6.8$ | $\mathbf{80.96} \pm 12.8$ | $72.36 \pm 5.5$ |
| walker2d-medium | $63.23 \pm 16.24$ | $82.5 \pm 3.6$ | $78.64 \pm 1.5$ | $\mathbf{79.08} \pm 1.5$ | $80.45 \pm 1.4$ | $78.96 \pm 1.1$ |
| walker2d-medium-expert | $98.96 \pm 15.98$ | $111.6 \pm 0.3$ | $107.05 \pm 2.3$ | $107.57 \pm 2.2$ | $\mathbf{108.99} \pm 0.7$ | $\mathbf{109.23} \pm 0.4$ |
| **MuJoCo-total** | 453.6 | 796.2 | **728.63** | **731.43** | 680.40 | **748.37** |

Therefore, we wonder what the performance looks like if we tune the hyperparameters for each reward function and for each dataset:

- `MuJoCo`: We use a batch size of 256. And we search for the best actor BC-regularization coefficient in $\{0.001, 0.01, 0.05, 0.1, 0.5\}$, and search for the best critic BC-regularization coefficient in $\{0, 0.001, 0.01, 0.1, 0.5\}$, for each MuJoCo dataset.

- `Adroit-pen` & `pen-cloned`: We search for the best actor BC-regularization coefficient in $\{0.001, 0.05, 0.01, 0.1\}$, and search for the best critic BC-regularization coefficient in $\{0, 0.001, 0.01, 0.1, 0.5\}$.

- `Antmaze`: We use a discount factor $\gamma = 0.99$, and set the learning rate of both actor and critic to be $3e^{-4}$. We then search for the best actor BC-regularization coefficient in $\{0.001, 0.002, 0.01, 0.02, 0.1, 0.2\}$, and search for the best critic BC-regularization coefficient in $\{0, 0.001, 0.01, 0.1\}$, for each antmaze dataset.

The chosen hyperparameters are reported in Table 9 and Table 10. Table 6 and 7 summarize the tuned results for OT reward and Segment-matching reward. We also include the oracle ReBRAC results for comparison. Instead of 10 seeds for experiments in the main body, we run 3 seeds for each experiment in Table 7. After hyperparameter tuning, we can see that the overall performance on Antmaze is much better than the untuned results for both OT and Segment-matching. Eventually, their total scores match each other. However, we still observe a large gap the comparing them with

---

[3]`https://github.com/tinkoff-ai/ReBRAC/tree/public-release/configs/rebrac`

the oracle results. On the one hand, this may suffer from a large variance when running on only 3 seeds. On the other hand, a more sophisticated IRL method than both OT and Segment-matching may be required to close this gap in future work.

Table 6: Normalized scores with ReBRAC as the downstream RL algorithm. Hyperparameters are **tuned** according to Table 9 and Table 10.

| Dataset | BC | ReBRAC (oracle) | OT | TemporalOT | Seg-match | Min-Dist |
|---|---|---|---|---|---|---|
| hopper-medium-replay | 29.81 ± 2.07 | 98.1 ± 5.3 | **98.57** ± 1.6 | **97.63** ± 2.1 | 91.86 ± 2.9 | **97.43** ± 1.4 |
| hopper-medium | 53.51 ± 1.76 | 102.0 ± 1.0 | **98.63** ± 1.0 | **98.43** ± 1.2 | 96.48 ± 3.4 | 95.09 ± 7.2 |
| hopper-medium-expert | 52.30 ± 4.01 | 107.0 ± 6.4 | **109.12** ± 2.0 | 108.04 ± 3.2 | 109.18 ± 2.1 | 107.94 ± 3.3 |
| halfcheetah-medium-replay | 35.66 ± 2.33 | 51.0 ± 0.8 | 46.19 ± 0.4 | **48.30** ± 0.7 | 47.43 ± 0.9 | 48.27 ± 0.9 |
| halfcheetah-medium | 42.40 ± 0.19 | 65.6 ± 1.0 | 47.28 ± 0.3 | **50.79** ± 0.7 | 44.49 ± 1.2 | **49.87** ± 0.7 |
| halfcheetah-medium-expert | 55.95 ± 7.35 | 101.1 ± 5.2 | **92.17** ± 1.2 | 92.71 ± 1.7 | 91.87 ± 1.6 | 92.45 ± 2.6 |
| walker2d-medium-replay | 21.80 ± 10.15 | 77.3 ± 7.9 | 71.24 ± 6.1 | 72.61 ± 7.2 | **84.57** ± 10.2 | 77.04 ± 4.0 |
| walker2d-medium | 63.23 ± 16.24 | 82.5 ± 3.6 | **80.53** ± 1.1 | 81.71 ± 0.9 | 83.99 ± 0.4 | 80.77 ± 1.0 |
| walker2d-medium-expert | 98.96 ± 15.98 | 111.6 ± 0.3 | **108.72** ± 0.5 | 108.88 ± 0.5 | 108.99 ± 0.7 | 109.25 ± 0.6 |
| **MuJoCo-total** | 453.6 | 796.2 | **752.46** | 759.11 | 758.85 | 758.11 |

Table 7: Normalized scores with ReBRAC as downstream RL algorithm. ReBRAC hyperparameters are tuned for datasets presented in this table, as well as for both OT reward and Segment-matching reward.

| Dataset | ReBRAC (oracle) | OT | Seg-match |
|---|---|---|---|
| pen-human | 103.5 ± 14.1 | **83.03** ± 6.56 | 70.25 ± 14.71 |
| pen-cloned | 91.8 ± 21.7 | **69.11** ± 15.02 | 46.57 ± 7.60 |
| antmaze-umaze-play | 97.8 ± 1.0 | **94.25** ± 3.18 | 93.42 ± 2.78 |
| antmaze-umaze-diverse | 88.3 ± 13.0 | 49.83 ± 5.8 | **62.75** ± 6.4 |
| antmaze-medium-play | 84.0 ± 4.2 | 59.17 ± 7.3 | **81.00** ± 3.2 |
| antmaze-medium-diverse | 76.3 ± 13.5 | 33.17 ± 25.6 | **43.08** ± 12.5 |
| antmaze-large-play | 60.4 ± 26.1 | **61.58** ± 6.1 | 42.75 ± 31.4 |
| antmaze-large-diverse | 54.4 ± 25.1 | **32.33** ± 14.6 | 33.25 ± 24.0 |
| **Tuned-total** | 656.5 | **482.47** | 473.07 |

Table 8: Normalized scores with TD3+BC as downstream RL algorithm.

| Dataset | TD3+BC (oracle) | OT | TemporalOT | Seg-match | Min-Dist |
|---|---|---|---|---|---|
| hopper-medium-replay | 74.94 ± 22.6 | **89.63** ± 6.4 | **92.11** ± 6.0 | 84.02 ± 18.6 | 34.65 ± 10.7 |
| hopper-medium | 67.04 ± 7.4 | 83.56 ± 4.9 | 83.84 ± 5.7 | **91.05** ± 4.6 | **87.06** ± 4.6 |
| hopper-medium-expert | 104.62 ± 7.8 | 102.30 ± 3.9 | 100.94 ± 6.4 | **107.83** ± 8.3 | **109.69** ± 3.8 |
| halfcheetah-medium-replay | 46.47 ± 0.6 | 39.20 ± 1.4 | **40.76** ± 0.6 | 40.73 ± 2.6 | **41.28** ± 0.6 |
| halfcheetah-medium | 50.97 ± 0.4 | **42.89** ± 0.4 | 44.18 ± 0.3 | 45.00 ± 0.3 | **44.90** ± 0.3 |
| halfcheetah-medium-expert | 81.69 ± 6.9 | 86.50 ± 2.8 | **91.95** ± 1.8 | 86.21 ± 3.8 | **92.74** ± 1.7 |
| walker2d-medium-replay | 85.00 ± 6.9 | 59.03 ± 21.3 | 46.96 ± 27.8 | **85.55** ± 5.0 | 61.32 ± 23.8 |
| walker2d-medium | 77.39 ± 24.9 | **72.44** ± 18.8 | **73.19** ± 21.7 | 17.10 ± 16.9 | 72.37 ± 10.2 |
| walker2d-medium-expert | 94.81 ± 32.2 | 55.61 ± 47.8 | **73.32** ± 40.4 | 41.86 ± 34.5 | 57.86 ± 33.2 |
| **MuJoCo-total** | 682.93 | **631.16** | 647.26 | 599.35 | 601.87 |
| pen-human | 1.74 ± 6.9 | -1.20 ± 2.9 | -1.46 ± 3.0 | **0.70** ± 5.2 | -0.69 ± 3.9 |
| pen-cloned | 5.10 ± 7.8 | 2.71 ± 8.2 | 3.57 ± 9.7 | 7.57 ± 8.8 | **8.13** ± 8.5 |
| door-human | -0.30 ± 0.1 | -0.34 ± 0.0 | -0.33 ± 0.0 | -0.33 ± 0.0 | -0.33 ± 0.0 |
| door-cloned | -0.34 ± 0.0 | -0.34 ± 0.0 | -0.34 ± 0.0 | -0.34 ± 0.0 | -0.34 ± 0.0 |
| hammer-human | 0.82 ± 0.4 | 0.85 ± 0.3 | 0.83 ± 0.3 | 0.91 ± 0.3 | **0.97** ± 0.4 |
| hammer-cloned | 0.26 ± 0.0 | **0.57** ± 0.6 | 0.33 ± 0.1 | 0.52 ± 0.4 | 0.46 ± 0.5 |
| relocate-human | -0.30 ± 0.0 | -0.29 ± 0.0 | -0.29 ± 0.0 | -0.30 ± 0.0 | -0.30 ± 0.0 |
| relocate-cloned | -0.30 ± 0.0 | -0.30 ± 0.0 | -0.30 ± 0.0 | -0.30 ± 0.0 | -0.30 ± 0.0 |
| **Adroit-total** | 6.68 | 1.66 | 2.00 | **8.43** | 7.61 |
| antmaze-umaze-play | 73.75 ± 36.1 | 47.50 ± 37.3 | 52.75 ± 43.2 | **75.50** ± 32.0 | 51.75 ± 39.5 |
| antmaze-umaze-diverse | 12.75 ± 17.2 | 26.00 ± 26.3 | 32.25 ± 28.3 | **44.75** ± 29.5 | 25.00 ± 30.8 |
| antmaze-medium-play | 0.75 ± 2.2 | **1.00** ± 1.9 | 0.75 ± 1.8 | 0.25 ± 0.8 | **1.00** ± 2.5 |
| antmaze-medium-diverse | 0.25 ± 0.8 | 0.75 ± 2.2 | **4.00** ± 7.5 | 1.00 ± 2.4 | 2.25 ± 4.0 |
| antmaze-large-play | 0.00 ± 0.0 | **0.00** ± 0.0 | **0.00** ± 0.0 | **0.00** ± 0.0 | **0.00** ± 0.0 |
| antmaze-large-diverse | 0.00 ± 0.0 | **0.00** ± 0.0 | **0.00** ± 0.0 | **0.00** ± 0.0 | **0.00** ± 0.0 |
| **Antmaze-total** | 87.50 | 75.25 | 89.75 | **121.50** | 80.00 |

**Evaluations with TD3+BC**  Prior to the ReBRAC, TD3+BC (Fujimoto & Gu, 2021) was first introduced as a minimalist solution to the core challenge of offline RL, known as *extrapolation error* (Levine et al., 2020). Under the general framework of behavior-regularized actor-critic approach, TD3+BC does not constrain the Q target but only regularizes the actor training objective of online TD3 by adding a behavior cloning loss. One can roughly consider TD3+BC as ReBRAC with the

critic regularization coefficient being 0 and the actor regularization coefficient being 1. In addition, TD3+BC does not employ tricks involved in ReBRAC, e.g., larger networks, larger batch size, and layer normalization (Tarasov et al., 2023a). As a result of such a reduction to an untuned primitive ReBRAC, TD3+BC cannot effectively learn from the IRL rewards and suffers from large performance variances for `walker2d` environment and catastrophically fails on `Adroit` and `Antmaze` domains, as we show in Table 8.

Table 9: Tuned regularization coefficients of actor and critic in ReBRAC algorithms for different reward functions. In each tuple, the first value is for the actor, and the second is for the critic.

| Dataset | OT | TemporalOT | Seg-match | Min-Dist |
|---|---|---|---|---|
| hopper-medium-replay | (0.01, 0.5) | (0.01, 0.5) | (0.1, 0.001) | (0.01, 0.5) |
| hopper-medium | (0.01, 0.1) | (0.01, 0) | (0.05, 0.5) | (0.01, 0.5) |
| hopper-medium-expert | (0.05, 0.1) | (0.05, 0.5) | (0.5, 0.001) | (0.1, 0) |
| halfcheetah-medium-replay | (0.001, 0.5) | (0.001, 0.5) | (0.01, 0.5) | (0.001, 0.1) |
| halfcheetah-medium | (0.001, 0.5) | (0.001, 0.5) | (0.05, 0.001) | (0.001, 0.1) |
| halfcheetah-medium-expert | (0.01, 0.001) | (0.01, 0) | (0.05, 0.5) | (0.01, 0) |
| walker2d-medium-replay | (0.1, 0.5) | (0.05, 0.5) | (0.05, 0.5) | (0.01, 0.1) |
| walker2d-medium | (0.05, 0.5) | (0.01, 0.5) | (0.05, 0.5) | (0.01, 0.1) |
| walker2d-medium-expert | (0.05, 0.1) | (0.01, 0.1) | (0.01, 0.01) | (0.01, 0.001) |

Table 10: Tuned regularization coefficients of actor and critic in ReBRAC algorithms for different reward functions. In each tuple, the first value is for the actor, and the second is for the critic.

| Dataset | OT | Seg-match |
|---|---|---|
| pen-human | (0.01, 0.001) | (0.1, 0.01) |
| pen-cloned | (0.05, 0.5) | (0.1, 0.1) |
| antmaze-umaze-play | (0.01, 0.001) | (0.01, 0.001) |
| antmaze-umaze-diverse | (0.1, 0.01) | (0.01, 0.1) |
| antmaze-medium-play | (0.001, 0.001) | (0.01, 0.001) |
| antmaze-medium-diverse | (0.002, 0.1) | (0.01, 0.1) |
| antmaze-large-play | (0.001, 0) | (0.002, 0.1) |
| antmaze-large-diverse | (0.001, 0.001) | (0.002, 0.001) |

## A.4 MULTI-EXPERT GENERALIZATION

We vary $K \in \{1, 5, 10, 20\}$ on the D4RL benchmark, using IQL as the downstream learner. Figure 5 reports the aggregated normalized scores for each domain, with error bars representing the accumulated standard deviations from each domain's datasets.

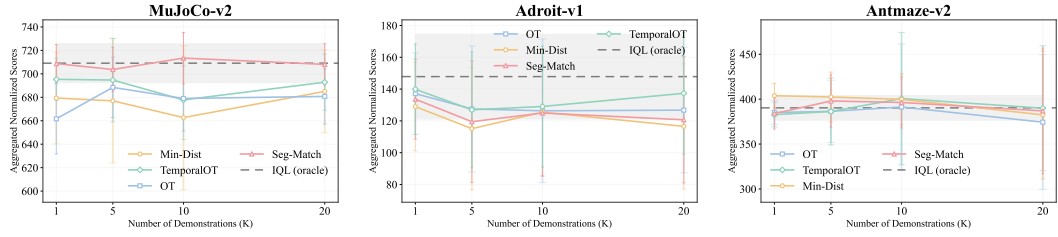

Figure 5: Multi-expert ablation results for **MuJoCo** (left), **Adroit** (middle) and **Antmaze** (right).

## A.5 COMPARISON WITH MORE RELATED WORK

We compare the performance of several methods discussed in Section 6 on MuJoCo datasets. Literature along this line of work has conducted extensive empirical analysis. Table 11 gathers the existing results from the literature under the same problem setup, including Luo et al. (2023) and Liu et al. (2023).

Note that simple heuristics like SQIL and UDS, which assign 0-1 rewards to the unlabeled dataset, do not work well in this challenging setup where limited expert demonstrations are available. Although CLUE seems to achieve the highest total score, we emphasize the following points, which make its result less prominent. First, CLUE uses an SAC expert to collect demonstrations for *all* MuJoCo datasets, while other work simply uses the best episode in each dataset. Second, CLUE requires

Table 11: Comparison of SQIL, UDS, IQ-Learn, ORIL, SMODICE, and CLUE on D4RL locomotion benchmarks.

| Dataset | SQIL | UDS | IQ-Learn | ORIL | SMODICE | CLUE |
|---|---|---|---|---|---|---|
| halfcheetah-medium | $24.3 \pm 2.7$ | $42.4 \pm 0.3$ | $21.7 \pm 1.5$ | $56.8 \pm 1.2$ | $42.4 \pm 0.6$ | $45.6 \pm 0.3$ |
| halfcheetah-medium-replay | $43.9 \pm 1.0$ | $37.9 \pm 2.4$ | $7.7 \pm 1.6$ | $46.2 \pm 1.1$ | $38.3 \pm 2.0$ | $43.5 \pm 0.5$ |
| halfcheetah-medium-expert | $6.7 \pm 1.2$ | $63.0 \pm 5.7$ | $2.0 \pm 0.4$ | $48.7 \pm 2.4$ | $80.9 \pm 2.3$ | $90.0 \pm 2.4$ |
| hopper-medium | $66.9 \pm 5.1$ | $54.5 \pm 3.0$ | $29.6 \pm 5.6$ | $96.3 \pm 0.9$ | $54.8 \pm 1.2$ | $78.3 \pm 5.4$ |
| hopper-medium-replay | $98.6 \pm 0.7$ | $49.3 \pm 2.7$ | $23.0 \pm 9.4$ | $56.7 \pm 12.9$ | $30.4 \pm 7.8$ | $94.3 \pm 6.0$ |
| hopper-medium-expert | $13.6 \pm 9.6$ | $53.9 \pm 2.5$ | $9.1 \pm 2.2$ | $25.1 \pm 12.8$ | $82.4 \pm 7.8$ | $96.5 \pm 14.7$ |
| walker2d-medium | $51.9 \pm 11.7$ | $68.9 \pm 6.2$ | $5.7 \pm 4.0$ | $20.4 \pm 13.6$ | $67.8 \pm 6.0$ | $80.7 \pm 1.5$ |
| walker2d-medium-replay | $42.3 \pm 5.8$ | $17.7 \pm 9.6$ | $17.0 \pm 7.6$ | $71.8 \pm 9.6$ | $49.7 \pm 4.6$ | $76.3 \pm 2.8$ |
| walker2d-medium-expert | $18.8 \pm 13.1$ | $107.5 \pm 1.7$ | $7.7 \pm 2.4$ | $11.6 \pm 14.7$ | $94.8 \pm 11.1$ | $109.3 \pm 2.1$ |
| **MuJoCo-total** | 367.0 | 494.9 | 123.5 | 434.6 | 541.5 | 714.5 |

| Dataset | IQL (oracle) | OT | TemporalOT | Seg-match | Min-Dist |
|---|---|---|---|---|---|
| hopper-medium-replay | $94.67 \pm 8.2$ | $\mathbf{95.03} \pm 1.3$ | $96.05 \pm 1.0$ | $\mathbf{97.10} \pm 1.0$ | $\mathbf{96.17} \pm 1.3$ |
| hopper-medium | $63.51 \pm 4.7$ | $\mathbf{75.16} \pm 4.5$ | $\mathbf{73.78} \pm 3.8$ | $\mathbf{77.52} \pm 5.0$ | $71.59 \pm 3.7$ |
| hopper-medium-expert | $103.60 \pm 9.0$ | $\mathbf{108.12} \pm 3.2$ | $\mathbf{107.58} \pm 4.6$ | $\mathbf{109.14} \pm 2.7$ | $103.09 \pm 20.1$ |
| halfcheetah-medium-replay | $43.13 \pm 1.6$ | $39.01 \pm 2.0$ | $37.07 \pm 2.4$ | $38.46 \pm 2.6$ | $\mathbf{42.75} \pm 1.1$ |
| halfcheetah-medium | $47.49 \pm 0.3$ | $\mathbf{43.28} \pm 0.7$ | $42.56 \pm 0.5$ | $42.76 \pm 0.3$ | $\mathbf{45.11} \pm 0.6$ |
| halfcheetah-medium-expert | $91.06 \pm 2.2$ | $\mathbf{92.08} \pm 1.3$ | $58.51 \pm 6.7$ | $\mathbf{91.44} \pm 1.5$ | $\mathbf{92.56} \pm 1.4$ |
| walker2d-medium-replay | $73.79 \pm 8.9$ | $\mathbf{55.09} \pm 22.0$ | $\mathbf{56.72} \pm 14.7$ | $52.16 \pm 16.2$ | $49.97 \pm 24.4$ |
| walker2d-medium | $80.66 \pm 4.0$ | $\mathbf{79.65} \pm 0.7$ | $\mathbf{77.78} \pm 3.4$ | $72.50 \pm 4.8$ | $\mathbf{80.17} \pm 0.5$ |
| walker2d-medium-expert | $111.06 \pm 0.2$ | $\mathbf{110.76} \pm 0.1$ | $110.14 \pm 0.1$ | $109.47 \pm 0.2$ | $\mathbf{110.85} \pm 0.2$ |
| **MuJoCo-total** | 708.96 | **698.18** | 660.19 | **690.54** | **692.27** |

more than one expert trajectory to train representations (Table 8 Liu et al. (2023)). Last, CLUE tunes and chooses the best temperature constant in the exponential squashing (Table 6 Liu et al. (2023)), while we do not tune and stick to one value throughout. Therefore, despite the simplicity, Seg-Match performs arguably the best among these methods on MuJoCo benchmarks.

## B  ONLINE RL EVALUATIONS

Table 12: Normalized scores for online RL on MetaWorld with DrQ-v2 as the downstream RL algorithm. DrQ-v2 (Oracle) denotes performance obtained from training with ground-truth *sparse* task reward.

| Dataset | DrQ-v2 (Oracle) | OT | TemporalOT | Seg-Match |
|---|---|---|---|---|
| window-open-v2 | $85.6 \pm 12.2$ | $15.75 \pm 8.5$ | $21.25 \pm 5.4$ | $\mathbf{22.75} \pm 4.7$ |
| door-open-v2 | $0.0 \pm 0.0$ | $73.05 \pm 12.2$ | $75.20 \pm 13.8$ | $\mathbf{79.35} \pm 3.6$ |
| door-lock-v2 | $86.2 \pm 12.4$ | $43.80 \pm 36.0$ | $\mathbf{67.10} \pm 7.1$ | $62.30 \pm 5.9$ |
| push-v2 | $1.0 \pm 0.7$ | $12.55 \pm 3.5$ | $\mathbf{17.12} \pm 2.3$ | $15.45 \pm 3.6$ |
| stick-push-v2 | $0.0 \pm 0.0$ | $\mathbf{96.95} \pm 2.1$ | $98.19 \pm 1.7$ | $97.80 \pm 1.3$ |
| basketball-v3 | $0.0 \pm 0.0$ | $73.90 \pm 35.6$ | $\mathbf{94.70} \pm 2.5$ | $85.85 \pm 4.6$ |
| button-press-topdown-v2 | $14.0 \pm 18.5$ | $\mathbf{65.10} \pm 6.2$ | $64.00 \pm 6.6$ | $65.65 \pm 5.1$ |
| hand-insert-v2 | $0.8 \pm 1.6$ | $8.30 \pm 2.8$ | $\mathbf{9.25} \pm 2.9$ | $8.85 \pm 3.0$ |
| lever-pull-v2 | $0.0 \pm 0.0$ | $2.50 \pm 1.4$ | $\mathbf{2.80} \pm 1.2$ | $1.50 \pm 1.3$ |
| **Metaworld-total** | 188.6 | 391.90 | **449.61** | **439.50** |

### B.1  REWARD POST-PROCESSING

Unlike in the offline setting, global reward normalization based on known return bounds is not feasible in the online case, as new trajectories are collected continuously and the maximum or minimum achievable returns are unknown a priori. To address this, we adopt the same reward scaling procedure used in Temporal OT (Fu et al., 2024). Specifically, we compute the sum of absolute rewards during the first episode,

$$\text{rewards\_sum} = \sum_t |r_t|$$

and define a fixed scaling coefficient:

$$\text{sinkhorn\_reward\_scale} = \frac{\text{reward\_scale\_factor}}{\text{rewards\_sum}}.$$

where we fix `reward_scale_factor = 10` for all experiments. This scaling factor is applied uniformly to all subsequent rewards during training, ensuring stable reward magnitudes while preserving the relative structure of the signal. For Segment-matching, we apply Gaussian smoothing with a std of 0.5 and a kernel radius of $T/2$ throughout.

**Existence of multiple experts**  In addition, online experiments assume two expert trajectories. In this case, rewards are computed independently for each expert demonstration. The set of rewards yielding the highest total cumulative reward is selected.

### B.2  HYPERPARAMETER SETTINGS

We adopt the same hyperparameter settings used in Fu et al. (2024) for all methods unless otherwise stated. All policies are trained using the DrQ-v2 algorithm, with vision-based observations and a total training horizon of 1 million environment steps. Table 13 summarizes the key parameters. Since Fu et al. (2024) only provides the expert demonstrations for `basketball-v3` and `door-open-v2` in their codebase, we use their code and the expert policies they provide to collect two expert trajectories for each of the rest of the tasks. However, we notice that the expert policies may sometimes generate suboptimal trajectories, and the performance cannot match what is reported in their paper. Despite this, we make sure that for each task, the expert demonstrations are fixed for all four reward functions for fair comparison.

### B.3  METAWORLD TASKS

We evaluate our methods on nine tasks from the MetaWorld benchmark (Yu et al., 2020), following the setup in Fu et al. (2024). Each task involves a vision-based robotic manipulation objective with fixed episode lengths. Specifically, the `basketball-v3` and `lever-pull-v2` tasks have horizon length 175, while all other tasks are capped at 125 steps. Task descriptions and goal specifications are provided below.

- **Basketball**: Grasp the ball and move it above the rim.
- **Button-press**: Press down a red button.
- **Door-lock**: Rotate the door lock knob to a target angle.
- **Door-open**: Open the door to a specified position.
- **Hand-insert**: Insert a brown block into a hole.
- **Lever-pull**: Pull the lever to a target height.
- **Push**: Push a red cylinder to a goal location on the table.
- **Stick-push**: Use a blue stick to push a bottle to the goal.
- **Window-open**: Slide the window to a designated open position.

Table 13: Hyperparameters for DrQ-v2 online evaluations on MetaWorld.

|  | Hyperparameter | Value |
| --- | --- | --- |
| Training | Total environment steps | 1e6 |
|  | Evaluation frequency | Every 5e4 steps |
|  | Evaluation episodes | 10 |
|  | Batch size | 512 |
|  | Discount factor $\gamma$ | 0.9 |
|  | Target network update rate $\tau$ | 0.005 |
|  | Learning rate | 1e-4 |
| Network Architecture | Actor hidden layers | (1024, 1024, 1024) |
|  | Critic hidden layers | (1024, 1024, 1024) |
|  | Embedding dimension | 50 |
|  | CNN channels | (32, 32, 32, 32) |
|  | CNN kernel sizes | (3, 3, 3, 3) |
|  | CNN strides | (2, 1, 1, 1) |
|  | CNN padding | VALID |
| DrQ-specific Settings | Buffer size | 1.5e5 |
|  | Action repeat | 2 |
|  | Frame stack | 3 |
|  | Image resolution | $84 \times 84 \times 3$ |
| Reward Labeling | Number of expert demonstrations | 2 |
|  | Context length $k_c$ | 3 |
|  | Temporal mask width $k_m$ (TemporalOT) | 10 |
|  | Segment window size (Seg-window) | 10 |
| Computation | GPU model | NVIDIA A100 8358 80GB |
|  | Parallel tasks per GPU | 5 |
|  | CPU workers per task | 2 |
|  | Memory per task | 16 GB |
|  | Approximate average execution time | 10 hours |

## C   FURTHER DISCUSSION ON REWARD COMPUTATION

### C.1   COMPUTATIONAL COMPLEXITY

We analyze the cost of computing reward labels for the various reward-labeling algorithms for a single non-expert trajectory of length $T$. To simplify the analysis, we assume $T_e = T$. Let $d$ denote the dimension of the state.

**OT**   The original OT method requires computing pairwise distances between all agent and expert states, forming a cost matrix of size $T \times T$. Each entry involves a $d$-dimensional comparison, resulting in $\mathcal{O}(T^2 d)$ time for cost matrix construction. Solving the entropic OT problem (e.g., via Sinkhorn iterations) typically adds no worse asymptotic cost, so the overall complexity remains $\mathcal{O}(T^2 d)$.

**TemporalOT**   If we ignore the cost of computing the context-aware cost matrix, the actual implementation of the temporal mask in the Temporal OT paper does not add additional computational cost on top of OT. TemporalOT sets all non-masked costs to a very large number, so that this forces the optimal coupling to assign probability mass only to expert states within the masking window. In terms of computing the context-aware cost matrix, the worst complexity is $\mathcal{O}(k_c T^2)$, where $k_c$ is the context length. Therefore, the total complexity is $\mathcal{O}((k_c + d)T^2)$.

**Seg-Match**   In contrast, the Segment-Match method assigns each agent state $s_t$ to a single non-overlapping expert segment of fixed size, and computes its minimum distance to that segment. Since the number of comparisons per state is constant, the total cost scales as $\mathcal{O}(Td)$.

**Min-Dist**   The Min-Dist method requires finding, for each non-expert state, its nearest neighbor in the entire expert trajectory. A brute-force implementation incurs $\mathcal{O}(T^2 d)$ cost. However, when $d$ is small, we may pre-build a kd-tree over expert states and reduce the query time to $\mathcal{O}(\log T)$, leading to a total complexity of $\mathcal{O}(T \log T + Td)$.

In practice, we do observe significant increases in GPU time spent on OT reward computation, as the number of expert demonstrations increases. For example, IQL with JAX implementation takes **20 minutes** for training 1 million steps on the Antmaze-large-play dataset. When the number of expert demonstrations is 20, OT with JAX implementation requires **48 minutes** for labeling this dataset. And TemporalOT with JAX needs **58 minutes**. Thus, the downstream RL algorithm is no longer the bottleneck of computation. This again emphasizes the significance of simplifying or eliminating the OT optimization.

### C.2   GENERALIZED SIMPLE REWARD FORMULATION

We describe a general framework for constructing proximity-based reward functions through temporally indexed windows over expert trajectories. This formulation abstracts over temporal alignment strategies by allowing flexible parameterization of comparison windows, and provides the foundation for the specific heuristics introduced in Section 4.

Let $\tau = (s_1, \ldots, s_T)$ be a non-expert trajectory, and $\tau^e = (s_1^e, \ldots, s_{T_e}^e)$ an expert trajectory. For each timestep $t \in \{1, \ldots, T\}$, a reward is assigned by comparing the agent state $s_t$ to a window of expert states $\mathcal{W}(t) \subseteq \tau^e$:

$$r_t = - \min_{s^e \in \mathcal{W}(t)} \mathrm{Dist}(s_t, s^e),$$

where $\mathrm{Dist}(\cdot, \cdot)$ denotes a chosen distance metric (e.g., cosine or Euclidean).

We define each window via three non-negative integers $a, b, c \in \mathbb{N}_0$. Here, $b$ is a stride factor that maps agent time to expert time, and $a$, $c$ control the backward and forward extent of the window, respectively. The expert window at timestep $t$ is given by:

$$\mathcal{W}(t) = \left\{ s_j^e \mid j \in [\lfloor bt \rfloor - a, \ \lfloor bt \rfloor + c] \cap [1, T_e] \cap \mathbb{Z} \right\}.$$

If $T > T_e$, the index $\lfloor bt \rfloor + c$ may exceed the expert horizon. In such cases, we default to the last available expert state:

$$\mathcal{W}(t) := \{s_{T_e}^e\}, \quad r_t := -\mathrm{Dist}(s_t, s_{T_e}^e), \quad \text{for } t > T_e.$$

The reward functions introduced in Section 4 arise as specific instances of this formulation.

**Minimum-Distance reward** corresponds to setting $b = 0$, so that the index $\lfloor bt \rfloor$ remains constant for all $t$. Choosing $a$ and $c$ large enough to span the full expert trajectory (e.g., $a = 1 - \lfloor bt \rfloor$, $c = T_e - \lfloor bt \rfloor$) yields $\mathcal{W}(t) = \tau^e$ for all $t$, recovering the full-pointwise comparison used in Min-Dist.

**Segment-Matching reward** is obtained by choosing $b = \frac{T_e}{T}$, and defining $a$ and $c$ to match the partitioning procedure described in Section 4. Specifically, the expert trajectory is divided into $T$ contiguous, approximately equal-length segments. The $t$-th agent state is compared against expert states in the corresponding segment $\Gamma_t$, which can be represented as a window centered at $\lfloor bt \rfloor$ with asymmetrical bounds determined by the residue $T_e \bmod T$.

This structure provides a principled basis for defining interpretable reward heuristics with temporal flexibility and aligns with the indexing and assumptions used throughout the method section.

# D   PROOF OF THEOREM 1

Intuitively, $J(\pi)$ can be interpreted as the average proximity to the expert trajectory in Wasserstein space when following $\pi$. The larger $J(\pi)$ is, the more closely $\pi$ follows the expert trajectory. Thus, Theorem 1 tells that the optimal policy learned by Min-Dist rewards is sufficient to rival the trajectory-matching ability of the optimal OT policy up to the bound. When the bound is small, then the simple Min-Dist algorithm does as well as OT both empirically and theoretically. We point out several scenarios where this bound is small in Appendix D.

- **Low-quality data regime**: Given a low-quality dataset, the marginal state-action distribution of the behavior policy $\pi_\beta$ concentrates around low-quality transitions whose proximity rewards $\mathcal{R}(s,a)$ are small. Thus, the first term of the bound becomes small. The second term is also likely to be small, as $d^{\pi^*_{min}}$ concentrates around expert transitions and thus their proximity rewards are high. Therefore, the bound becomes small. Indeed, Table 2 shows that OT only achieves 160.99 on `MuJoCo-medium-replay` datasets, while Min-Dist achieves 200.16.

- **High-quality data regime**: In this case, both $d^{\pi_\beta}$ and $d^{\pi^*_{min}}$ should concentrate around the expert transitions. For near-expert states, it is very likely that the reward difference between $\mathcal{R}(s,a)$ and $r^{min}_{s,a}$ is close to zero. Therefore, the bound should be small in this case. In Table 2, OT (73.43) and Min-Dist (71.47) perform similarly on `Adroit-human` datasets.

- **Better offline RL algorithm**: A better algorithm can learn a policy from Min-Dist rewards that approximates expert behavior better. In this case, the third term of the bound becomes small. Considering the ReBRAC results in Figure 2, Min-Dist (758.11) indeed exhibits better performance than OT (752.46).

**Theorem 2** (Sufficiency Bound for Min-Dist Reward; full version). *Let $\mathcal{M} = (\mathcal{S}, \mathcal{A}, \mathcal{P}, \mathcal{R}, \gamma)$ be a finite-horizon deterministic MDP with $\gamma \in [0,1)$, where*

- *state-action space $\mathcal{S} \times \mathcal{A}$ is finite;*

- *transition dynamics $\mathcal{P}$ is known and deterministic;*

- *reward function $\mathcal{R}(s,a) := \mathbb{E}_{\tau \sim \mathcal{M}} \mathcal{R}_{ot}(s,a,\tau;\tau_e) \in (0,1]$ computes the stationary average OT reward over all possible trajectories $\tau \in \mathcal{M}$ containing $(s,a) \in \tau$, where $\mathcal{R}_{ot}$ is the OT reward labeling algorithm and $\tau_e$ is a fixed expert demonstration.*

*A reward-free offline dataset $\mathcal{D} = \{(s,a)\}$ is given and let the corresponding behavior policy to be $\pi_\beta$. Denote $\mathcal{D}_{ot} = \{(s,a,r^{ot}_{s,a})\}$ to be the dataset relabeled by OT rewards; and $\mathcal{D}_{min} = \{(s,a,r^{min}_{s,a})\}$ to be the dataset relabeled by Min-Dist rewards.*

*Assume that the optimal policy is learned through a generic constrained offline policy optimization problem:*

$$\pi^*_x := \arg\max_\pi \hat{J}_{\mathcal{D}_x}(\pi) - \frac{\alpha}{1-\gamma}\mathrm{Div}(\pi,\pi_\beta)$$

*where $x \in \{ot, min\}$; $\hat{J}_{\mathcal{D}_x}(\pi)$ is the average return of policy $\pi$ in the empirical MDP induced by dataset $\mathcal{D}_x$; Div measures divergence between distributions; and $\alpha$ controls the constraint strength.*

*Assume that the labeled OT rewards $r^{ot}_{s,a} \in \mathcal{D}_{ot}$ concentrates towards the mean with high probability $\geq 1 - \delta$:*

$$|r^{ot}_{s,a} - \mathcal{R}(s,a)| \leq \frac{C_{R,\delta}}{\sqrt{\mathcal{D}(s,a)}}$$

*where $C_{R,\delta}$ is a positive constant; and $\mathcal{D}(s,a)$ denotes the number of visits to $(s,a)$ in $\mathcal{D}$.*

*Define $\rho(s,a) = \frac{|\{(s,a,r) \in \mathcal{D}_{min} | r \neq r^{ot}_{s,a}\}|}{\mathcal{D}(s,a)}$ to be the ratio of reward mismatch between OT and Min-Dist labeling. Then, with high probability $\geq 1 - \delta$, $\pi^*_{ot}$ is no better than $\pi^*_{min}$ by the following bound:*

$$J(\pi^*_{ot}) - J(\pi^*_{min}) \leq \frac{1}{1-\gamma} \mathbb{E}_{d^{\pi_\beta}}\left[\rho(s,a) \cdot \left(\mathcal{R}(s,a) - r^{min}_{s,a}\right)\right]$$

$$- \frac{1}{1-\gamma} \mathbb{E}_{d^{\pi^*_{min}}}\left[\rho(s,a) \cdot \left(\mathcal{R}(s,a) - r^{min}_{s,a}\right)\right]$$

$$- \frac{\alpha}{1-\gamma} \mathrm{Div}(\pi^*_{min}, \pi_\beta) + \mathcal{O}\left(\frac{1}{1-\gamma}\right)$$

*where $J(\pi)$ is the average return of $\pi$ in the MDP $\mathcal{M}$; and $d^\pi(s,a)$ denotes the marginal state-action distribution of $\pi$.*

*Proof.* First, we want to get an upper and lower bound on $J(\pi) - J(\pi^\beta)$ for any $\pi$. By Theorem 4.1 in Yu et al. (2022), we obtain the following lower bound:

$$J(\pi) - J(\pi_\beta) = J(\pi) - \hat{J}(\pi) + \hat{J}(\pi) - \hat{J}(\pi_\beta) + \hat{J}(\pi_\beta) - J(\pi_\beta)$$

$$\geq -\frac{2\gamma C_{P,\delta}}{(1-\gamma)^2} \mathbb{E}_{s \sim d^\pi_{D_x}}(s)\left[\frac{\sqrt{|\mathcal{A}|}}{\sqrt{|D_x(s)|}}\sqrt{\mathrm{Div}(\pi,\pi_\beta)(s)} + 1\right]$$

$$- \frac{2C_{R,\delta}}{1-\gamma} \mathbb{E}_{s,a \sim d^\pi_{D_x}}\left[\frac{1 - \rho(s,a)}{\sqrt{|D_x(s,a)|}}\right]$$

$$- \frac{1}{1-\gamma} \mathbb{E}_{s,a \sim d^{\pi_\beta}_{D_x}}\left[\rho(s,a)\left(\mathcal{R}(s,a) - r^{s,a}_x\right)\right]$$

$$+ \frac{1}{1-\gamma} \mathbb{E}_{s,a \sim d^\pi_{D_x}}\left[\rho(s,a) \cdot \left(\mathcal{R}(s,a) - r^{s,a}_x\right)\right]$$

$$+ \frac{\alpha}{1-\gamma} \mathrm{Div}(\pi,\pi_\beta).$$

Since we assume known dynamics, the first term vanishes as it is only related to uncertainty of unknown dynamics. So, we have:

$$J(\pi) - J(\pi_\beta) = J(\pi) - \hat{J}(\pi) + \hat{J}(\pi) - \hat{J}(\pi_\beta) + \hat{J}(\pi_\beta) - J(\pi_\beta)$$

$$\geq -\frac{2C_{R,\delta}}{1-\gamma} \mathbb{E}_{s,a \sim d^\pi_{D_x}}\left[\frac{1 - \rho(s,a)}{\sqrt{|D_x(s,a)|}}\right]$$

$$- \frac{1}{1-\gamma} \mathbb{E}_{s,a \sim d^{\pi_\beta}_{D_x}}\left[\rho(s,a)\left(\mathcal{R}(s,a) - r^{s,a}_x\right)\right]$$

$$+ \frac{1}{1-\gamma} \mathbb{E}_{s,a \sim d^\pi_{D_x}}\left[\rho(s,a) \cdot \left(\mathcal{R}(s,a) - r^{s,a}_x\right)\right]$$

$$+ \frac{\alpha}{1-\gamma} \mathrm{Div}(\pi,\pi_\beta).$$

Moreover, note that:

$$\hat{J}(\pi) - \hat{J}(\pi_\beta) = \frac{1}{1-\gamma} \sum_{s,a} \left(d^\pi_\mathcal{D}(s)\,\pi(a \mid s) - d^{\pi_\beta}_\mathcal{D}(s)\,\pi_\beta(a \mid s)\right)\mathcal{R}(s,a)$$

$$\leq \frac{1}{1-\gamma} \sum_{s,a} d^\pi_\mathcal{D}(s)\,\pi(a \mid s)$$

$$= \frac{1}{1-\gamma}$$

where the inequality follows since $\mathcal{R} \leq 1$. Putting this into Theorem 4.1 in Yu et al. (2022), we get a similar upper bound:

$$J(\pi) - J(\pi_\beta) = J(\pi) - \hat{J}(\pi) + \hat{J}(\pi) - \hat{J}(\pi_\beta) + \hat{J}(\pi_\beta) - J(\pi_\beta)$$

$$\leq \quad \frac{2C_{R,\delta}}{1-\gamma} \mathbb{E}_{s,a \sim d^\pi_{D_x}} \left[ \frac{1 - \rho(s,a)}{\sqrt{|D_x(s,a)|}} \right]$$

$$+ \frac{1}{1-\gamma} \mathbb{E}_{s,a \sim d^{\pi_\beta}_{D_x}} \left[ \rho(s,a) \left( \mathcal{R}(s,a) - r^{s,a}_x \right) \right]$$

$$- \frac{1}{1-\gamma} \mathbb{E}_{s,a \sim d^\pi_{D_x}} \left[ \rho(s,a) \cdot \left( \mathcal{R}(s,a) - r^{s,a}_x \right) \right]$$

$$+ \frac{1}{1-\gamma}$$

Note that:

$$J(\pi^*_{ot}) - J(\pi^*_{min}) = J(\pi^*_{ot}) - J(\pi_\beta) - (J(\pi^*_{min}) - J(\pi_\beta))$$

Then, we want an upper bound on $J(\pi^*_{ot}) - J(\pi_\beta)$ and note that the reward mismatch $\rho(s,a) = 0$ for $r^{s,a}_{ot}$:

$$J(\pi^*_{ot}) - J(\pi_\beta) \leq \frac{2C_{R,\delta}}{1-\gamma} \mathbb{E}_{s,a \sim d^{\pi^*_{ot}}_D} \left[ \frac{1}{\sqrt{|D(s,a)|}} \right] + \frac{1}{1-\gamma}$$

Similarly, we have:

$$J(\pi_{min}) - J(\pi_\beta) \geq -\frac{2C_{R,\delta}}{1-\gamma} \mathbb{E}_{s,a \sim d^{\pi^*_{min}}_{D_x}} \left[ \frac{1 - \rho(s,a)}{\sqrt{|D_x(s,a)|}} \right]$$

$$- \frac{1}{1-\gamma} \mathbb{E}_{s,a \sim d^{\pi_\beta}_{D_x}} \left[ \rho(s,a) \left( \mathcal{R}(s,a) - r^{s,a}_{min} \right) \right]$$

$$+ \frac{1}{1-\gamma} \mathbb{E}_{s,a \sim d^{\pi^*_{min}}_{D_x}} \left[ \rho(s,a) \cdot \left( \mathcal{R}(s,a) - r^{s,a}_{min} \right) \right]$$

$$+ \frac{\alpha}{1-\gamma} \mathrm{Div}(\pi, \pi_\beta).$$

Putting together the two inequalities, we will have the final results. $\qquad \square$

