# OpenReview forum: "Is Optimal Transport Necessary for Inverse Reinforcement Learning?"
_ICLR.cc/2026/Conference — Submitted to ICLR 2026_

### Official Review · Reviewer_WRY6 · 2025-10-20

**Soundness:** 2
**Presentation:** 2
**Contribution:** 2
**Rating:** 4
**Confidence:** 3

**Summary:**

This paper argues that optimal transport is computationally expensive and unnecessary for IRL and proposes two simpler reward (minimum-distance reward and segment-matching reward) that are easier to obtain. The paper also provides theoretical guarantee of the proposed techique, particularly that the proposed simpler reward does not perform worse than the OT reward.

**Strengths:**

This paper proposes two novel reward design. Compared to the OT reward, the proposed reward is simpler to obtain and the authors provide theoretical guarantees that the proposed reward does not perform worse than the OT reward.

**Weaknesses:**

1. I find the setting is different from standard IRL. In standard IRL, the demonstrations only include expert trajectories and the reward and policy are updated iteratively. In this paper, the demonstrations include both expert and nonexpert trajectories and reward learning and policy learning are performed in a sequential fashion. I am wondering the benefits of doing in this way? Intuitively, this setting is more restrictive as it requires nonexpert trajectories and I suspect that there is an implicit assumption on the quality of the nonexpert trajectories. If the nonexpert trajectories are very bad, e.g., generated by a random policy, I do not think this contrastive reward can be much useful.

2. The proposed approaches just, in essence, match individual states. They ignore the causal relation between states in an MDP, i.e., next state depends on the current state. The causal relation between consecutive states within a trajectory is not utilized. In that sense, for the first approach, there is no need to collect trajectories, you can just sample some states from the stationary state distribution of the expert/non-expert policies.

**Questions:**

1. Can the authors elaborate the benefits of IRL in this paper compared to the standard IRL setting?

2. Can the authors discuss how to use the causal relation between consecutive states to learn a reward in this setting?

---

### Official Review · Reviewer_nizn · 2025-10-29

**Soundness:** 2
**Presentation:** 3
**Contribution:** 2
**Rating:** 2
**Confidence:** 4

**Summary:**

This paper questions the value of optimal transport (OT) in offline reinforcement learning, proposing that much simpler methods are equally effective. The authors introduce two alternatives: 1) Minimal-Distance Reward and 2) Segment-Matching Reward. Empirical results on the MuJoCo benchmark suggest that these proposed methods perform on par with recent OT-based approaches for IRL and offline RL.

**Strengths:**

1)	The motivation is clear, and the paper is well-written and easy to follow.
2)	The proposed approaches are conceptually simple and perform competitively with more complex adversarial OT methods.

**Weaknesses:**

1)	The empirical support for the proposed methods is inconsistent across benchmarks. On MuJoCo, Segment-Matching performs best, while on Antmaze, Minimal-Distance is superior. This variability limits the methods' applicability to more complex benchmarks, as it necessitates training and comparing both approaches to determine the best one.
2)	The paper lacks a discussion on why more complex min-max OT solutions are not more robust for IRL, resulting in only marginal performance gains compared to the much simpler Minimal-Distance method. Is the problem in Sinkhorn entropy regularization, estimation of Kantorovich potentials (for neural case)?
3)	No discussion on application of more robust neural OT approaches for IL. Thus, it is not clear whether proposed methods beat only Sinkhorn-like OT solvers.

**Questions:**

1)	Given the results, can the authors definitively claim that OT-based methods are not useful for offline RL/IRL? Should future works adopt Minimal-Distance and Segment-Matching rewards and consider min-max OT optimization obsolete? While Theorem 1 seems to support this, the empirical evidence is not uniformly conclusive.

2)	I would appreciate a discussion on the intuitive reasons for the underwhelming performance of min-max OT approaches (which only outperform by a maximum of 10-15%). Intuitively, incorporating temporal context for closeness comparison (as in TemporalOT and Segment-Matching) should always be superior to the temporal Minimal-Distance. However, this holds true only for MuJoCo. Why does this intuition fail for other benchmarks like Antmaze?

3)	Why more performant for offline RL is not taken, e.g DemoDICE? It would be beneficial to compare to those and would strengthen empirical part of the work.

4)	All baselines are based on Sinkhorn-regularized OT problem. Is it possible to compare proposed approach to more robust OT solvers based on neural nets? Consider DIOTM [1], Expectile-OT [2] as recent state-of-the art methods from neural OT field.

5)	From the author’s perspective, is OT in Offline RL / IRL has future research directions? Or it is better to focus on other approaches to  imitation learning?

[1] Choi et al. Improving Neural Optimal Transport via Displacement Interpolation, ICLR 2025

[2] Buzun et al. ENOT: Expectile Regularization for Fast and Accurate Training of Neural Optimal Transport, NeurIPS 2024

---

### Official Review · Reviewer_DYoV · 2025-10-31

**Soundness:** 3
**Presentation:** 3
**Contribution:** 2
**Rating:** 2
**Confidence:** 4

**Summary:**

The authors solve the offline imitation learning problem by proposing a simple alternative to the commonly used optimal transport. Authors showcase 2 simple heuristics for labeling trajectories based on distance similarity. The authors benchmark these reward labeling functions on the common dataset D4RL, showing similar performance to OT algorithms.

**Strengths:**

1. Simplicity of the method. The method itself does not require complex algorithms and is purely based on trajectory alignment via distance.
2. Temporal alignment. The method by design includes the property of temporal alignment, allowing trajectories to encode context information, which is crucial for obtaining a reward aligned with the goal.
3. Multiple datasets and mixture of experts evaluation. The authors evaluated their algorithm across multiple datasets and demonstrated its performance in a scenario with a mixture of experts.

**Weaknesses:**

1. **Incremental contribution:** The proposed method appears to offer only a modest extension of existing approaches and does not achieve state-of-the-art performance. This raises questions regarding the practical significance and broader applicability of the method.

2. **Incomplete comparison with prior work:** The authors claim superior performance in most cases; however, they do not include a comparison with [1], which also leverages a single expert trajectory and consistently achieves better results across the same benchmarks. This omission limits the strength of their empirical claims.

[1] Bobrin, Align Your Intents: Offline Imitation Learning via Optimal Transport. https://arxiv.org/pdf/2402.13037

**Questions:**

1. Could you please add comparisons with [1] and explain any benefits regarding your work besides simplicity?
2. Have you tried using thresholded Euclidean distance? i.e if the distance above d_max we set reward to -r_max.
3. How would cosine distance compare with embedded latent distance ||z_1 - z_2||_2 i.e if we learn a very simple autoencoder and use this instead of using pure Euclidean distance?


[1] Bobrin, Align Your Intents: Offline Imitation Learning via Optimal Transport. https://arxiv.org/pdf/2402.13037

---

### Official Review · Reviewer_eeZD · 2025-11-01

**Soundness:** 3
**Presentation:** 3
**Contribution:** 3
**Rating:** 4
**Confidence:** 3

**Summary:**

The paper looks into whether solving an OT coupling is truly necessary for IRL-style reward labeling. It revisits a standard OT reward and TemporalOT (OT with context windows and temporal masking), and proposes two optimization-free surrogates: Min-Dist (negative nearest distance from an agent state to the expert) and Seg-Match (negative nearest distance within the temporally corresponding expert segment). A complexity table and wall-clock measurements show that reward labeling can dominate runtime for OT variants in practice. The study compares all four labelers with IQL/ReBRAC/DrQ-v2 across D4RL and MetaWorld.

**Strengths:**

1. This work proposes two optimization-free reward surrogates (Min-Dist, Seg-Match) with clear computational advantages;
2. Provide a practical complexity and timing comparison for reward labeling;
3. Empirical study shows that simple distance-based surrogates are competitive or better on many standard setups.

**Weaknesses:**

1. The paper does not test harder cases where OT variants are known to be useful (e.g., large time warps, cross-domain state geometry, partial observability, goal shifts). As the authors themselves highlight, outcomes are highly sensitive to downstream RL hyperparameters (γ, BC regularizations). Hence, the claim “OT is unnecessary” should be re-scoped to the studied regimes. The paper already shows tuning can flip results (e.g., Seg-Match collapse under untuned regularization vs. recovery after tuning), underscoring the fragility of broad claims.
2. The comparison focuses on “vanilla” OT and TemporalOT; modern OT variants (unbalanced/partial OT for length mismatch, GW for structural alignment across spaces, sliced/mini-batch/low-rank OT for efficiency, time-regularized/soft-DTW couplings for large warps) are not compute-matched in the study. Without these, concluding “OT is unnecessary” feels under-substantiated beyond the specific instantiations tested.
3. Seg-Match presumes rough temporal alignment (or similar speeds/lengths); otherwise, distances inflate and rewards become biased. This is acknowledged and empirically manifested: with untuned downstream settings, Seg-Match can abruptly fail (e.g., halfcheetah-medium). A systematic length/speed-mismatch sweep would strengthen the claim.

**Questions:**

1. Temporal alignment is a critical concept in this work; you may need to explicitly discuss this in the work, specifically in the introduction.
2. Can you please clarify and provide more details for the selection method when multiple experts' trajectories are available?
3. Table 2 is actually in the Appendix; you may want to mention this.
3. It claims "In contrast, Seg-Match and TemporalOT can be improved with more expert demonstrations.". However, from K 10 to 20, there is a significant drop in the performance of Seg-Match, which makes the claims not fully convincing.

---

### Meta-Review · Area_Chair_Z3s9 · 2026-01-05

**Summary:**

This paper on inverse reinforcement learning got 4 negative answers and the authors did not answer any of their comments, hence the recommendation is a clear reject

**Reviewer Scores:**

N/A

---

### Decision · Program_Chairs · 2026-01-26

Reject